# The role of atmospheric rivers in the distribution of heavy precipitation events over North America

Sara M. Vallejo-Bernal[1,2], Frederik Wolf[1], Niklas Boers[1,3,4], Dominik Traxl[1], Norbert Marwan[1,2], and Jürgen Kurths[1,2]

[1]Research Domain IV - Complexity Science, Potsdam Institute for Climate Impact Research (PIK) – Member of the Leibniz Association, Potsdam, Germany
[2]Institute of Geoscience, University of Potsdam, Germany
[3]Earth System Modelling, School of Engineering & Design, Technical University of Munich, Germany
[4]Global Systems Institute and Department of Mathematics, University of Exeter, UK

**Correspondence:** Sara M. Vallejo-Bernal (vallejo.bernal@pik-potsdam.de)

**Abstract.** Atmospheric rivers (ARs) are filaments of extensive water vapor transport in the lower troposphere, that play a crucial role in the distribution of fresh water, but can also cause natural and economical damage by facilitating heavy precipitation. Here, we investigate the large-scale spatio-temporal synchronization patterns of heavy precipitation events (HPEs) over the western coast and the continental regions of North America (NA), during the period from 1979 to 2018. In particular, we utilize event synchronization and a complex network approach incorporating varying delays to examine the temporal evolution of spatial patterns of HPEs in the aftermath of land-falling ARs. For that, we employ the SIO-R1 catalog of ARs that landfall on the western coast of NA, ranked in terms of strength and persistence on an AR-intensity scale which varies from level AR1 to AR5, along with daily precipitation estimates from the ERA5 reanalysis with $0.25°$ spatial resolution. Our analysis reveals a cascade of synchronized HPEs, triggered by ARs of level AR3 or higher: on the first 3 days after an AR makes landfall, HPEs mostly occur and synchronize along the western coast of NA. In the subsequent days, moisture can be transported to central and eastern Canada and cause synchronized but delayed HPEs there. Furthermore, we assess the robustness of our findings by studying an additional AR detection method. Finally, analyzing the anomalies of integrated water vapor transport, geopotential height, upper-level meridional wind, and precipitation, we find atmospheric circulation patterns that are consistent with the spatio-temporal evolution of the synchronized HPEs. Understanding and revealing the effects of ARs in the precipitation patterns over NA will lead to a better understanding of inland HPEs and how changing climate dynamics may affect precipitation occurrence and consequent impacts in the context of a warming atmosphere.

## 1  Introduction

Atmospheric rivers (ARs) are channels of enhanced water vapor flux that transport moisture over thousands of kilometres, often from the tropics to the mid-latitudes of both hemispheres, where they can cause substantial precipitation following landfall (e.g. Zhu and Newell, 1998; Ralph and Dettinger, 2011; Baggett et al., 2017; Eiras-Barca et al., 2018; Mundhenk et al., 2018; Shields et al., 2018; Ralph et al., 2019; Payne et al., 2020; O'Brien et al., 2022). Located in the lower troposphere, ARs can persist from

several hours to several days, carrying as much water as the Amazon river (Newell et al., 1992). Therefore, they play a crucial role in the global water cycle (Neiman et al., 2008), the Arctic water influx (Baggett et al., 2016), and the occurrence of heavy precipitation events (HPEs) (Neiman et al., 2008; Krichak et al., 2015). However, intense and persistent ARs have also been associated with natural hazards such as extreme winds, floods, and landslides, and their respective economic losses (Ramseyer and Teale, 2021; Sharma and Déry, 2020; Corringham et al., 2019; Ralph et al., 2019; Waliser and Guan, 2017). Further motivated by their high environmental, social and economical impacts, occurrences of ARs have been studied intensively in the last decade, especially on the western coasts of North America (NA) and Europe (Smith et al., 2010; Dettinger et al., 2011; Newman et al., 2012; Lavers and Villarini, 2013; Warner et al., 2015; Krichak et al., 2015; Shields and Kiehl, 2016; Ramos et al., 2016; Baggett et al., 2017; Shields et al., 2018; Mundhenk et al., 2018; Eiras-Barca et al., 2018; Ralph et al., 2019; Lora et al., 2020; Guirguis et al., 2020; Eiras-Barca et al., 2021; Huang et al., 2021; O'Brien et al., 2022).

With this increased interest in understanding the dynamics, impacts and future evolution of ARs, a plethora of methodological approaches to identify and track these atmospheric features, both in space and time, has been proposed and multiple AR catalogs have been produced and made available to the public (Gershunov et al., 2017; Guan and Waliser, 2015; Prabhat et al., 2021; Pan and Lu, 2019; Traxl, 2022). This abundance of information has brought big challenges to the climate research community, as the climatological statistics of ARs have proven to be highly dependent on the identification method used (Huning et al., 2017), affecting in particular the resultant AR climatologies and the attribution of high-impact weather and climate events to ARs (Shields et al., 2018). As a collective effort to address these issues, the Atmospheric River Tracking Method Intercomparison Project (ARTMIP) has quantified and analyzed the uncertainties in AR science based on the choice of detection/tracking methodology (Shields et al., 2018; Rutz et al., 2019; O'Brien et al., 2020; O'Brien et al., 2022; Lora et al., 2020). The results achieved by the scientific community involved provide nowadays the guidelines for choosing the most appropriate algorithm for a given scientific question or region of interest.

As a consequence, novel and relevant topics of AR science have been studied, such as the initiation and evolution of ARs and their moisture sources (Guan and Waliser, 2019; Waliser and Guan, 2017; Rutz et al., 2014), and the foreseen response of ARs to a warmer or different climate (Gao et al., 2015; Hagos et al., 2016; Payne et al., 2020). Thanks to these contributions, the following key findings have been recently revealed to the climate scientific community: *i)* in the northern hemisphere, ARs usually originate in the mid-latitude ocean basins (Guan and Waliser, 2019) paired with extratropical cyclones (Zhang et al., 2019). *ii)* As they travel east, ARs accumulate and transport moisture, primarily to the western coasts of North America and Europe, where they facilitate precipitation and play a key role both in the fresh water supply and in the occurrence of HPEs. *iii)* In the context of ongoing climate change, a poleward shift of the land-falling location, together with an increase in the frequency and intensity of ARs, can be expected in the coming decades as a response to the higher water vapor content in a warmer atmosphere (Gao et al., 2016; Hagos et al., 2016; Payne et al., 2020). Moreover, compared with the present, ARs affecting middle and high elevations are expected to result in more liquid than solid precipitation, exacerbating the potential risk and severe impacts of natural hazards such as floods and landslides (Mahoney et al., 2018).

In light of the scientific knowledge that has been gained in recent decades about ARs, there has been an increasing effort in characterizing and predicting the landfall of ARs along the North American west coast by presenting comprehensive analyses

of their drivers and properties. However, the spatio-temporal synchronization patterns of heavy precipitation events (HPEs) induced by ARs have not yet been studied. Here, we understand spatio-temporal synchronization as a relation between pairs of precipitation time series measured at different locations, for which events in one time series are significantly followed or preceded by events in the other one. Such an assessment has led, among other findings (see e.g. Boers et al., 2013; Stolbova et al., 2014; Agarwal et al., 2019; Wolf et al., 2020b), to forecasting precipitation events in the Eastern Central Andes (Boers et al., 2014a) and identifying Rossby waves as one controlling mechanism of HPEs worldwide (Boers et al., 2019). In this light, it has not been examined to what extent ARs are accompanied by characteristic synchronization patterns of HPEs. Additionally, the lag-dependent spatial impact of ARs making landfall on the western coast of NA remains unrevealed. To elaborate on the issues explained and examined by the ARTMIP project (Shields et al., 2018; Rutz et al., 2019; O'Brien et al., 2022), in this study we address these specific research questions using two different catalogs with different AR-tracking schemes. Both catalogs considered here, the SIO-R1 product which was recently published by Gershunov et al. (2017) and a self-constructed one which is based on the IPART algorithm Xu et al. (2020); Traxl (2022), cover the period between 1979 and 2018. Based on the occurrence of ARs, we perform time series and complex network analyses evaluating the spatio-temporal correlation of HPEs and their relation to ARs. To interpret our results, we furthermore study the corresponding climatologies of different variables such as wind and geopotential height at 500 hPa.

The paper is structured as follows: first, we introduce the employed data sets and methods, in particular the characteristics of the two AR catalogs, the ERA5 reanalysis and the event synchronization (ES) and complex network techniques. Second, we conduct an ES-based assessment of the temporal correlation between land-falling ARs and HPEs for different lags. Having revealed different temporal scales at which AR-related HPEs occur, we set up two climate networks based on HPEs taking place at different lags. Finally, we study composite anomalies of integrated water vapor transport, geopotential height, wind, and precipitation for the times during which we identified features of synchronized HPEs and we discuss our findings in the context of the guiding climatology.

## 2 Data and Methods

### 2.1 Data sets

For our analyses, we utilize data from the ERA5 reanalysis (Hersbach et al., 2020; ECMWF, 2021). All ERA5 data sets are available on a longitude-latitude grid with a spatial resolution of $0.25° \times 0.25°$. We construct daily estimates for integrated water vapor transport (IVT), geopotential height at 500 hPa, wind at 650 hPa, and precipitation by considering the daily mean of the hourly data sets for the period between 1979 and 2018. To examine the synchronization of HPEs, we especially consider the 95[th] percentile thresholds of the daily precipitation estimates. Only days exceeding 1 mm of total precipitation, which we refer to as *wet days*, are used for computing the percentiles.

Recent studies have revealed the biases present in the ERA5 reanalysis, especially for precipitation estimations. Disagreements on the number of wet days, the co-occurrence of precipitation events and the precipitation intensity were identified, along with a consistent pattern of decreasing agreement with increasing intensity of events, independently of the season (Rivoire et al.,

2021). Larger differences were found over western NA, were ERA5 has additional difficulties in detecting and estimating oro-
graphic precipitation events (Adhikari and Behrangi, 2022). However, and despite of these biases, we use ERA5 estimates for
our analyses to maintain the consistency between the variables and their agreement with the large-scale circulation patterns.

In addition to the ERA5 reanalysis data set, we use the SIO-R1 catalog of ARs by Gershunov et al. (2017). It includes
ARs land-falling on the western coast of NA and was constructed using Lagrangian backtracking of high values of two vari-
ables, namely the vertically integrated horizontal vapor transport (IVT), and the vertically integrated water vapor (IWV), on a
longitude-latitude grid with a resolution of $2.5° \times 2.5°$. The catalog features a 6-hourly time series indicating whether an AR
has been active, the grid cells covered by the AR, and the IVT over the grid cell along the coast where the AR made landfall.
We transform the 6-hourly catalog into a daily one by considering each day with at least one (of the four) 6-hourly time step
with an active AR as an *AR-day*. Approximately one-third of all days of the analysis period are AR-days (at least one AR active
somewhere in the spatial domain covered by the catalog). These days are distributed relatively equally over the different years
but are strongly seasonally clustered in the boreal autumn and winter.

Furthermore, we create an additional catalog of ARs with features similar to the SIO-R1 catalog but using the Image-
Processing-based Atmospheric River Tracking (IPART) algorithm (Xu et al., 2020). As opposed to conventional detection
methods that rely on thresholding of IVT and/or IWV fields (for instance the detection algorithm of the SIO-R1 catalog), IPART
implements the detection task from a spatio-temporal scale perspective and is, therefore, free from magnitude thresholds. The
advantage of IPART's approach is that it negates the implicit assumption of thresholding approaches that the atmospheric
moisture level stays unchanged throughout the analysis period. As input to the IPART algorithm, we use IVT-fields of the
ERA5 reanalysis data set re-gridded to a spatial resolution of $0.75° \times 0.75°$ and a temporal resolution of 6 hours. The parameters
passed to the different steps of the IPART algorithm are summarized in Table A1 in appendix A (Traxl, 2022). We transform
the 6-hourly product into a daily one in the same manner as described for the SIO-R1 catalog.

To separate the impact of rather weak ARs from strong ARs, we rank AR events of both catalogs in terms of strength and
persistence on the AR-intensity scale proposed by Ralph et al. (2019) and using the notation in Eiras-Barca et al. (2021). As a
result, each AR event is assigned a level that increases from AR1 to AR5 and we run our analysis repeatedly, excluding ARs
from lower levels.

## 2.2 Event Synchronization (ES) between land-falling ARs and HPEs

Event synchronization (ES), as initially introduced by Quiroga et al. (2002), is a temporal correlation measure that quantifies
the co-variability of events in a pair of time series. While it has been originally proposed for analyzing spike trains in electroen-
cephalographic time series, ES is nowadays an established tool employed to construct climate networks (Malik et al., 2012;
Boers et al., 2013, 2019; Ozturk et al., 2019; Wolf et al., 2020a).

To define ES, let the sequence $\{t_l^\mu\}_{\mu=1,...,n_l}$ denote the time series of ARs of level $l$ or higher, making landfall at the western
coast of NA, and let the sequence $\{t_i^\nu\}_{\nu=1,...,n_i}$ denote the time series of HPEs observed at grid cell $i$. The total number of
land-falling ARs of level $l$ or higher and the total number of HPEs at grid cell $i$ are denoted by $n_l$ and $n_i$ respectively. We say
that the HPE $\nu$, observed at location $i$, at time $t_i^\nu$, is *synchronized* with the *preceding* AR event $\mu$, of level $l$ or higher, which

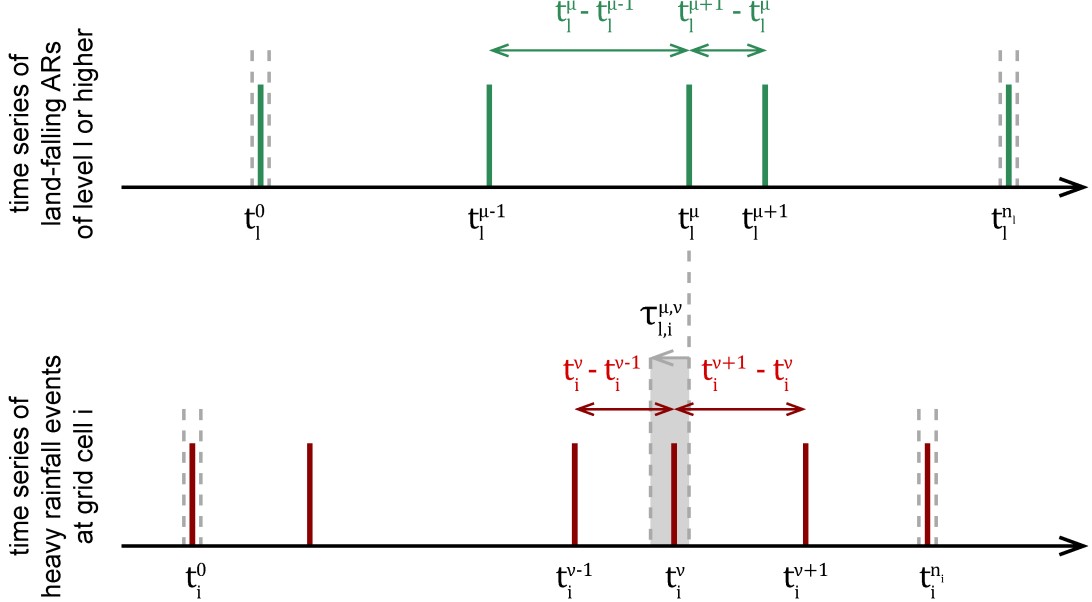

**Figure 1.** Schematic sketch illustrating event synchronization (ES). Shown is a pair of time series and an exemplary computation of the dynamical delay $\tau_{l,i}^{\mu;\nu}$. Additionally, the first and last events which are not considered for ES are marked by vertical grey lines.

made landfall at time $t_l^\mu$, if and only if their temporal delay, $t_i^\nu - t_l^\mu \geq 0$, does not exceed the *dynamical delay*

$$\tau_{l,i}^{\mu,\nu} = \frac{1}{2} \min \left\{ t_l^\mu - t_l^{\mu-1}, t_l^{\mu+1} - t_l^\mu, t_i^\nu - t_i^{\nu-1}, t_i^{\nu+1} - t_i^\nu \right\}. \tag{1}$$

This adaptive lag allows us to consider events in more densely and more sparsely occupied parts of the time series in an automated manner, in contrast to the classical lead-lag approach that only allows one lead or lag for the entire time series. As the first and last event of each time series has no preceding or subsequent event, we exclude them from our computations and only consider $\mu = 2, 3, ..., n_l - 1$ and $\nu = 2, 3, ..., n_i - 1$. See Fig. 1 for an illustration of the computation of the dynamical delay $\tau_{l,i}^{\mu,\nu}$.

To avoid a collapse of the dynamical delay to $\tau_{l,i}^{\mu;\nu} = \frac{1}{2}$ timestep due to sequences of consecutive events, we only consider the first event of all event sequences (cluster-corrected ES) (Boers et al., 2019; Wolf and Donner, 2021). Additionally, we can limit the dynamical delay $\tau_{l,i}^{\mu,\nu}$ by a minimal value $\tau_{\min}$ to consider a minimum lag between synchronized events, and also by a maximal value $\tau_{\max}$ to prevent an unrealistically large temporal delay between synchronized events.

Then, the synchronization condition reads:

$$S_{l,i}^{\mu,\nu} = \begin{cases} 1, & \text{if } 0 \leq t_i^\nu - t_l^\mu \leq \tau_{l,i}^{\mu,\nu} \text{ and } \tau_{\min} \leq \tau_{l,i}^{\mu,\nu} \leq \tau_{\max}, \\ 0, & \text{otherwise}, \end{cases} \tag{2}$$

and for each grid cell $i$, we define

$$ES_{l,i} = \sum_{\mu=2}^{n_l-1} \sum_{\nu=2}^{n_i-1} S_{l,i}^{\mu,\nu}, \tag{3}$$

as the total number of HPEs that can be uniquely associated with a *preceding* land-falling AR in a time-resolved manner, within an interval of at least $\tau_{\min}$ days and no more than $\tau_{\max}$ days.

Finally, we analyze the statistical significance of each empirical value $ES_{l,i}$, by means of a null model that incorporates $1{,}000$ surrogate pairs of time series of land-falling ARs and HPEs, preserving the original number of events $n_l$ and $n_i$ respectively, but destroying a potential correlation structure. We calculate the value of $ES_{l,i}$ from the surrogates to estimate an empirical probability distribution, that we then use to infer the significance level of our $ES_{l,i}$ value (Boers et al., 2019). We say that $ES_{l,i}$ is significant at the level $\alpha = 1 - \rho$ if

$$\Theta(ES_{l,i} - ET_\rho(n_l, n_i)) = 1, \tag{4}$$

where $\Theta$ denotes the Heaviside function, and $ET_\rho(n_l, n_i)$ is the $\rho-th$ percentile of the surrogate test distribution for $ES_{i,j}$.

## 2.3 Identification of ARs highly synchronized with a specific region

One particular advantage of ES is that we can use it to identify the temporal ordering and the time delay of synchronized events. From the synchronization condition (see Eq. (2)), note that $S_{l,i}^{\mu,\nu} = 1$ if and only if the land-falling AR event $\mu$ precedes and is synchronized with the HPEs event $\nu$ observed at location $i$. In that case, the time delay between this pair of uniquely associated events is $d_{l,i}^{\mu,\nu} = t_i^\nu - t_l^\mu$. If we use Eq. (4) to only consider the grid cells where the synchronization between land-falling ARs and HPEs is significant, then for a region of interest $R$, we can define

$$ES_{\mu \to R}^\rho = \sum_{i \in R} \sum_{\nu=2}^{n_i-1} S_{l,i}^{\mu,\nu} \Theta(ES_{l,i} - ET_\rho(n_l, n_i)), \tag{5}$$

as the total number of HPEs within $R$ that were preceded and uniquely associated with the land-falling AR $\mu$ at the significance level $\alpha = 1 - \rho$, during the time window $[\tau_{\min}, \tau_{\max}]$ (Boers et al., 2019). Based on this definition, we can retrieve the time delays between the AR event $\mu$ and the significantly synchronized HPEs,

$$D_{\mu \to R} := \{d_{l,i}^{\mu,\nu} \mid i \in R \,\wedge\, S_{l,i}^{\mu,\nu} = 1 \,\wedge\, \Theta(ES_{l,i} - ET_\rho(n_l, n_i)) = 1\}, \tag{6}$$

to define and calculate the *typical synchronization delay* between the AR event $\mu$ and the region of interest $R$ as the mode of $D_{\mu \to R}$ (if there are multiple modes, we take the smallest one).

We use this framework to select the ARs of level $l$ or higher with the largest number of significantly synchronized HPEs within the region of interest $R$ and to identify the time points that are then used to compute the composite anomalies of integrated water vapor transport, geopotential height, upper-level meridional wind, and precipitation to be shown in our results.

Even more, we can use Eq. (5) to identify ARs that were not synchronized with HPEs in the region of interest ($ES_{\mu \to R}^\rho = 0$). Based on a precedence analysis, we can also identify the dates when HPEs occurred in the region of interest without any AR

making landfall on the coast during the previous 12 days (the selection of this preceding time window comes after one of our results). These time points are used to compute composite anomalies of the aforementioned climatological variables, in order to reveal the synoptic conditions that differentiate ARs distributing synchronized HPEs into the region of interest.

## 2.4 Event Synchronization (ES) between HPEs at different locations

We are also interested in investigating if there is a directed synchronization pattern between HPEs at different locations in the aftermath of land-falling ARs. Adapting the definition of ES in section 2.2, we consider two HPEs $\nu$ and $\varphi$ in time series describing observations made at grid cells $i$ and $j$ at times $t_i^\nu$ and $t_j^\varphi$ as *synchronized* if and only if their temporal delay, $t_j^\varphi - t_i^\nu$, does not exceed the *dynamical delay*

$$\tau_{i,j}^{\nu,\varphi} = \frac{1}{2} \min \left\{ t_i^\nu - t_i^{\nu-1}, t_i^{\nu+1} - t_i^\nu, t_j^\varphi - t_j^{\varphi-1}, t_j^{\varphi+1} - t_j^\varphi \right\}. \tag{7}$$

Again, sequences of consecutive events are counted as single events (cluster-corrected ES), and the first and last event of each time series are discarded, i.e. $\nu = 2, 3, ..., n_i - 1$ and $\varphi = 2, 3, ..., n_j - 1$, where $n_i$ and $n_j$ denote the total number of HPEs at grid cells $i$ and $j$ respectively.

In this case, when the event at $i$ happens before the event at $j$, the synchronization condition reads:

$$S_{i,j}^{\nu,\varphi} = \begin{cases} 1, & \text{if } 0 < t_j^\varphi - t_i^\nu \leq \tau_{i,j}^{\nu,\varphi} \text{ and } \tau_{\min} \leq t_j^\varphi - t_i^\nu \leq \tau_{\max}, \\ 0, & \text{otherwise,} \end{cases} \tag{8}$$

Note the subtle but important difference with Eq. (2), we do not include events that occur at the very same time step at different locations, since we cannot determine their temporal ordering.

We define the directed event synchronization from $i$ to $j$ as

$$ES_{i,j} = \sum_{\nu=2}^{n_i-1} \sum_{\varphi=2}^{n_j-1} S_{i,j}^{\nu,\varphi}, \tag{9}$$

which is the total number of synchronized events where an event at $i$ precedes an event at $j$ by at least $\tau_{\min}$ days and no more than $\tau_{\max}$ days.

The reverse time direction is given by

$$ES_{j,i} = \sum_{\varphi=2}^{n_j-1} \sum_{\nu=2}^{n_i-1} S_{j,i}^{\varphi,\nu}, \tag{10}$$

resulting in the asymmetric matrix **ES**, which we use for setting up climate networks as described in the following section.

## 2.5 Climate networks

Functional networks are defined as graphs where nodes represent the elements of a complex system and edges represent the interaction between them. In functional networks, edges are placed between nodes in accordance with some statistical

similarity, regardless of whether the nodes are physically connected or not. A climate network, as introduced in former work by Tsonis and Roebber (2004), Tsonis et al. (2006), and Donges et al. (2009a), is a functional network whose nodes are identified with climatological time series, typically measured at specific spatial locations or grid cells, and whose edges account for a significant and strong correlation between the respective time series. Recently, the climate network approach has attracted much attention after being successfully applied to reveal novel insights into the dynamics of the Earth's climate system, over different spatiotemporal scales (Tsonis and Swanson, 2008; Yamasaki et al., 2008; Donges et al., 2009b, 2011; Malik et al., 2012; Steinhaeuser et al., 2012; Boers et al., 2014b, 2015; Agarwal et al., 2019; Boers et al., 2019; Messier et al., 2019; Wolf et al., 2020b).

A network consists of $N$ nodes connected by $e$ edges. The topology of such a network is commonly encoded in the adjacency matrix $\mathbf{A}$, with elements $a_{i,j}$ indicating if nodes $i$ and $j$ are connected. In this study, we construct climate networks based on ES to assess the spatiotemporal correlation structure of HPEs in NA, and to unravel possibly non-linear and long-ranged climatic linkages associated with the landfall of ARs on the western coast of NA. We identify the nodes of the network with the gridded time series of the ERA5 reanalysis data and connect the nodes based on their statistical association evaluated by ES. To transform the daily ERA5 precipitation estimates to an event time series, we threshold the time series of wet days of each grid cell at the 95th percentile. Subsequently, we apply the cluster-corrected ES (Boers et al., 2014a; Wolf et al., 2020a) with a lower and upper threshold of the dynamical delay which is specified in the respective sections. We use the resulting synchronization matrix $\mathbf{ES}$ to place a directed link pointing from grid cell $i$ to grid cell $j$ if HPEs at $i$ precede and are synchronized with HPEs at $j$.

Note that, by construction, we typically have different numbers of events at different grid cells, since we calculate the percentile of the wet days, not of the entire time series. The number of events affects the measured values of ES, therefore, to ensure that the edges placed between nodes are statistically significant, we apply a locally tailored significance testing scheme. For each pair of grid cells $(i, j)$, we set up a null model using $1,000$ surrogate pairs of time series preserving the respective numbers of events $n_i$ and $n_l$. We assume that the events at each location occur independently according to a uniform random distribution and compute the values $ES_{i,j}$ and $ES_{j,i}$ for each surrogate pair of time series, obtaining two empirical distribution functions. We then connect nodes in the network (by setting $a_{i,j} = 1$ in the adjacency matrix) if $ES_{i,j}$ exceeds the 99.5th percentile of the respective surrogate test distribution (Boers et al., 2019):

$$a_{i,j} = \Theta\left(ES_{i,j} - ET_{0.995}(n_i, n_j)\right) - \delta_{i,j}, \tag{11}$$

where $\Theta$ denotes the Heaviside function, $ET_{0.995}(n_i, n_j)$ is the 99.5th percentile of the surrogate test distribution for $ES_{i,j}$ and Kronecker's delta $\delta$ is used to exclude self loops.

As a remark, we want to emphasize that considering the temporal ordering of the events to calculate ES (see Eqs. (9) and (10)), results in a *directed climate network*, for which the adjacency matrix is not symmetric (in contrast to undirected networks). Moreover, since we also do not consider edge weights, the adjacency matrix is binary: $a_{i,j} = 1$ if an edge points from node $j$ to node $i$, $a_{i,j} = 0$ otherwise.

This methodology of combining ES and complex networks is based on the idea that ARs influence the way HPEs synchronize at different locations, and that these effects are contained in the internal structure of the climate network, which can be accessed by appropriate complex network measures. In many cases, climate networks are a powerful alternative to more traditional approaches based on eigenvalue techniques (e.g., PCA) (Boers et al., 2013). However, the methodology employed here has been specifically developed to analyze time series of climatological extreme events, for which PCA-like methods are not

applicable due to the binary-like structure and the non-Gaussian distributions of the data (Malik et al., 2012; Boers et al., 2013, 2014a, c, b, 2015, 2019; Stolbova et al., 2014).

    To calculate the number of nodes to which node $i$ is connected, we compute the in-degree $k_i^{in}$ (out-degree $k_i^{out}$) as the total number of links pointing to (from) grid cell $i$. To aggregate both measures and to highlight regions of predominately outgoing (or incoming) connections, we define the *network divergence* as

$$235 \quad d_i = k_i^{out} - k_i^{in} = \sum_j a_{i,j} - \sum_j a_{j,i}, \tag{12}$$

    with positive (negative) values of $d_i$ indicating sources (sinks) of the network: HPEs in these locations are followed (preceded) by HPEs in other locations.

## 3   Results and discussion

### 3.1   HPEs synchronized with strong ARs

As a first step, we investigate where HPEs occur synchronized to land-falling ARs and at which lags. For that, we employ ES and evaluate the synchronization between the AR time series and the time series of HPEs at each grid point in the study area. To obtain the latter one, we threshold the precipitation time series during wet days at the respective 95th percentile. For the former one, we consider ARs from the SIO-R1 catalog making landfall on the North American west coast at a latitude $\geq 47.5°$N. Initially, we included all ARs but additional analyses showed that our results are predominantly caused by ARs that landfall

north of $47.5°$N (see appendix A, Fig. A1). Moreover, we want to emphasize that our results can be reproduced using an alternative AR catalog based on the IPART algorithm (also featured in appendix A, see Fig. A2). To separate the impact of rather weak ARs from strong ARs, we differentiate between AR levels (classification based on Ralph et al. (2019) using the notation in Eiras-Barca et al. (2021)) and run the analysis repeatedly, excluding ARs from lower levels.

    Figure 2 shows the grid points whose time series of HPEs are significantly synchronized with the AR time series, given a

250 particular parameter setting of ES and ARs of level AR3 or higher. First, note that when $\tau_{\min} = 0$ (left column), large areas close to the western coast of NA show significant correlations. This pattern is caused by HPEs on the coast that are directly triggered by ARs and is not affected by increasing $\tau_{\max}$. If events occur in close succession, then a higher possible maximal delay will often not be taken into account and, therefore, the pattern does not change visibly. This strong synchronization between land-falling ARs and HPEs on the western coast of NA was expected, as it has already been implied by findings of

255 previous studies (Neiman et al., 2008; Gershunov et al., 2017; Waliser and Guan, 2017; Ralph et al., 2019), and serves as

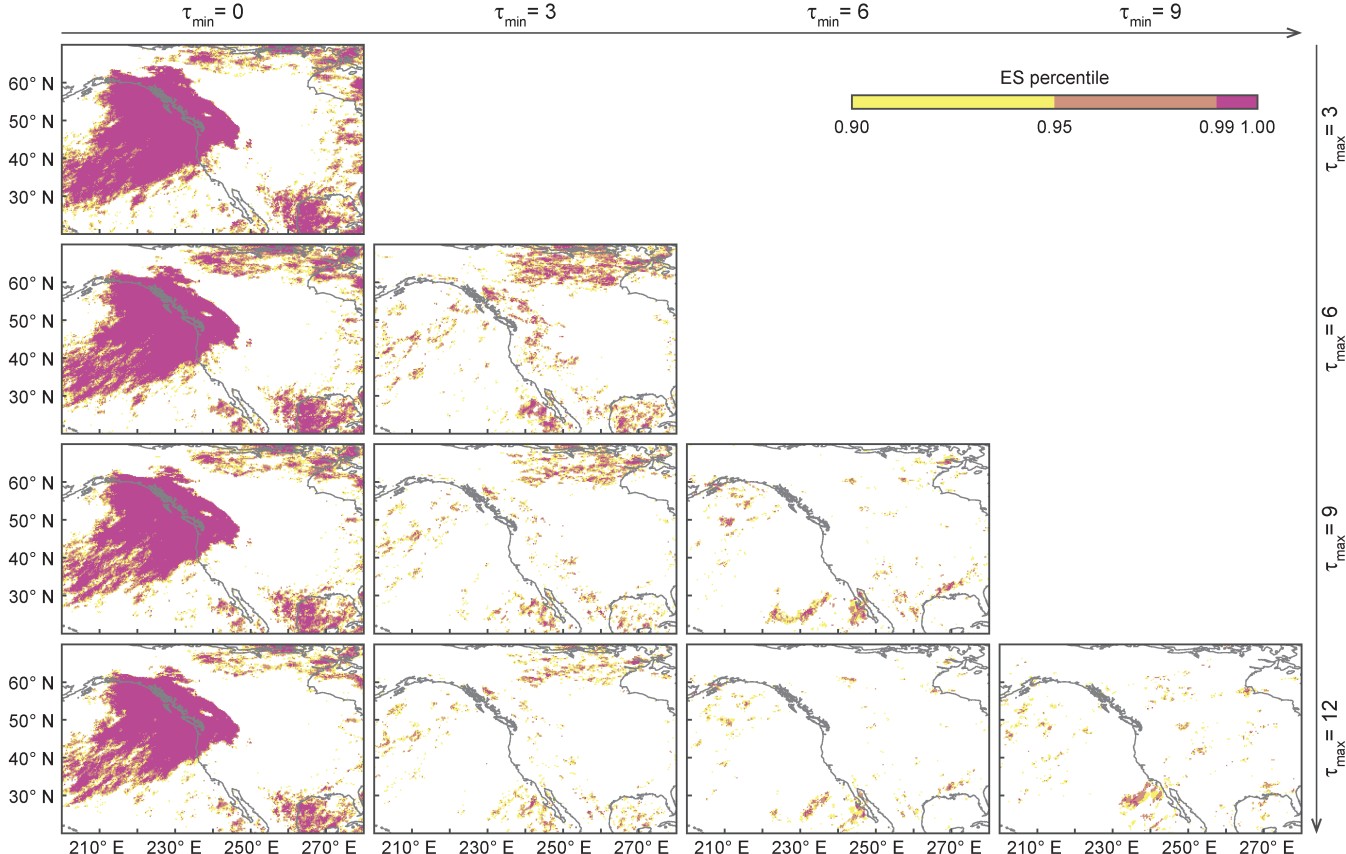

**Figure 2.** Event synchronization (ES) between ARs making landfall on the western coast of NA and HPEs. We use the SIO-R1 catalog of land-falling ARs but only consider ARs of level AR3 or higher with land-falling latitude north of $47.5°$N. Different values of $\tau_{min}$ and $\tau_{max}$ are considered to calculate ES in each panel: $\tau_{min}$ increases from left to right and $\tau_{max}$ from top to bottom. Note the irregularly spaced color bar: Yellow indicates high synchronization between HPEs and ARs of level AR3 or higher, at a significance level of $\alpha = 0.1$. Orange indicates high synchronization at a significance level of $\alpha = 0.05$ (and ES percentile $> 0.95$). Pink indicates high synchronization at a significance level of $\alpha = 0.01$ (and ES percentile $> 0.99$).

a proof of concept for our methodology. Excluding ARs of the lower levels AR1 and AR2 does not change this result (see appendix A, Fig. A3). Therefore, we have selected ARs of level AR3 or higher for our analysis.

When $\tau_{min} = 3$ (second column), the synchronization close to the coast decreases, as most HPEs occurs on the first days after an AR makes landfall. Additionally, most ARs do not persist longer than 3 days (Gershunov et al., 2017). The remaining
260 synchronized events are likely associated with ARs of the higher levels which have a longer persistence. Additionally, we observe a patch of synchronized events in central and eastern Canada. This pattern is strongest when $\tau_{min} = 3$ and $\tau_{max} = 6$ and stands out to a smaller extent up to $\tau_{max} = 12$. For elevated values of $\tau_{min}$ (third and fourth columns), the synchronization pattern completely vanishes.

This result implies that in central and eastern Canada, HPEs occurs synchronized (but lagged) to ARs making landfall on the western coast of NA. We, therefore, suspect that moisture that has been transported to the North American west coast by an AR can be channeled to central and eastern Canada and also cause HPEs there, as hinted in previous studies by Rutz et al. (2014), Waliser and Guan (2017), and Guan and Waliser (2019).

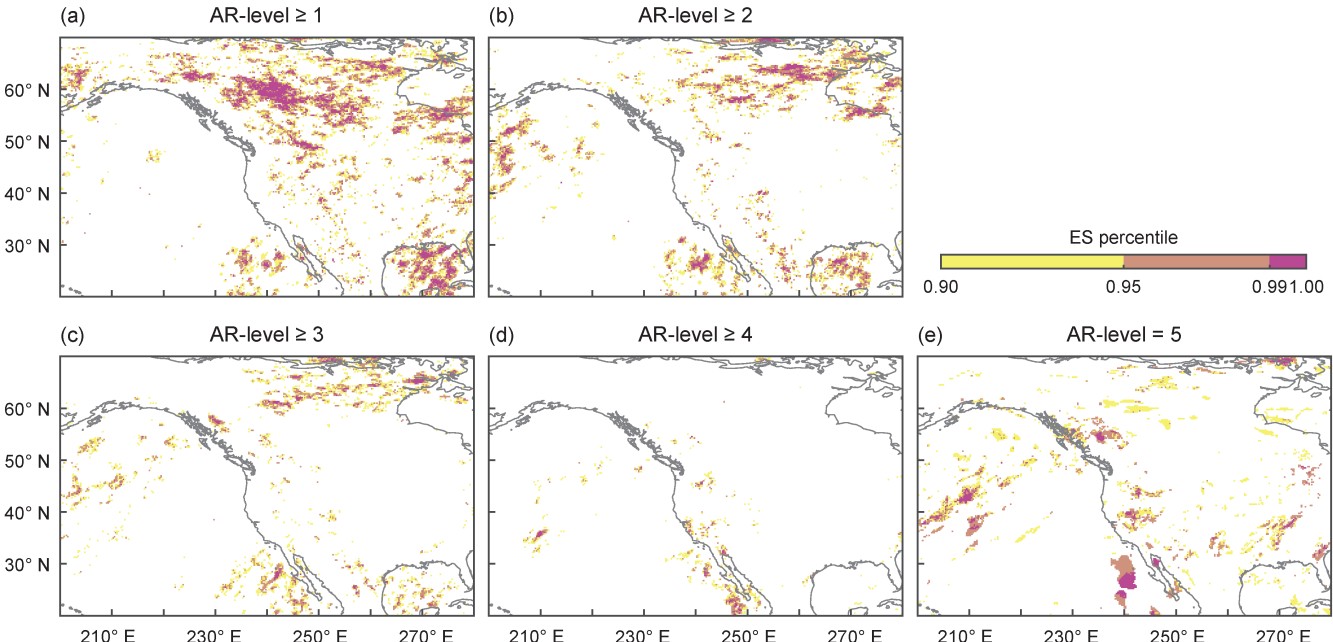

**Figure 3.** Event synchronization (ES) between ARs making landfall on the western coast of NA and HPEs. We use the SIO-R1 catalog of land-falling ARs but only consider ARs making landfall north of $47.5°$N. ES is calculated with $\tau_{min} = 3$ and $\tau_{max} = 12$. From (a) to (e) the lower limit of the considered AR level increases: (a) ARs of level AR1 and higher e.g. all ARs, (b) ARs of level AR2 and higher, rest accordingly. Color bar as in Fig. 2.

## 3.2 Synchronization across AR strength

Using ES, we have identified a region of synchronized HPEs in central and eastern Canada, as explained in the previous section. To further evaluate how the results depend on the selected AR-level criteria, we step-wise exclude ARs of the lower levels from the analysis, as shown in Fig. 3. Doing this, we reduce the number of events in the AR time series, which heavily affects the outcome of ES. As a result, we find that when we consider all ARs, the signal in central and eastern Canada is present but is accompanied by a more prominent synchronization pattern right next to the western coast of NA (Fig. 2). When we discard low-level ARs, the pattern next to the coast is filtered out and when we take into account ARs of level AR3 or higher, just the signal in central and eastern Canada is left. Only examining ARs of levels AR4 and AR5 leads to a vanishing of the synchronization in central and eastern Canada, although precipitation anomalies show HPEs in that region for composites

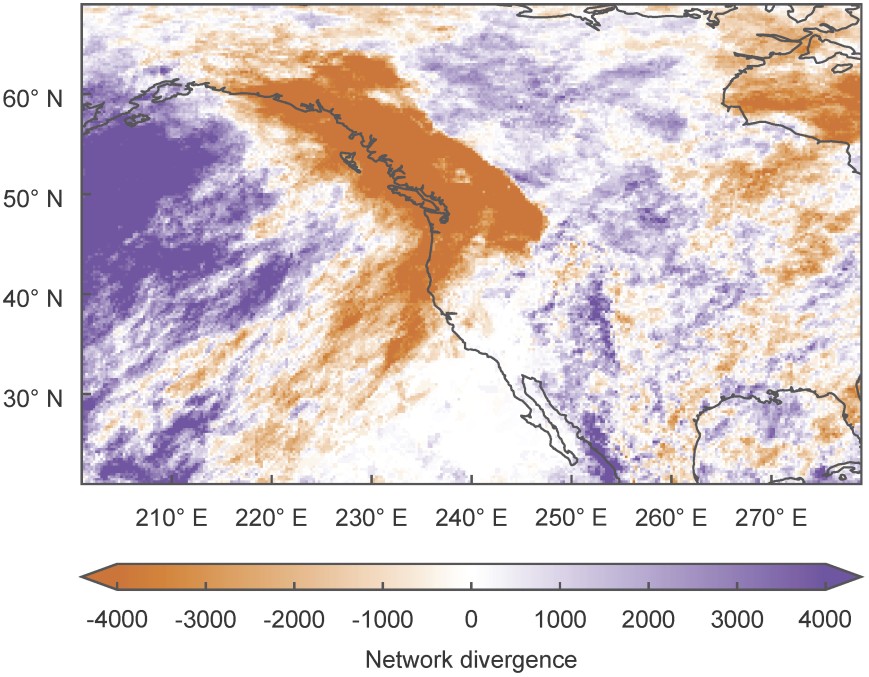

**Figure 4.** Network divergence based on event synchronization (ES) between HPEs at different locations during AR-days. We calculate ES with $\tau_{\min} = 0$ and $\tau_{\max} = 3$, and we only consider HPEs that occurred from $0$ to $3$ days after an AR of level AR3 or higher makes landfall north of $47.5°$N according to the SIO-R1 catalog. In the network, only nodes with significantly directed event synchronization are connected (see Sec. 2.5 for more details). Purple (orange) colors indicate regions with positive (negative) divergence, i.e. nodes with more outgoing (incoming) connections.

based on the days after such strong ARs (see appendix A, Fig. A6). This is a result of reducing the number of events in the AR time series. Very intense ARs are rare and when we only consider them, the AR time series contains very few events. In other words, the AR time series get too sparse and the ES scores are not significant any more, although HPEs might be always caused by such ARs but likely not *just* by them.

### 3.3 Network analysis of HPEs during AR-days

In the previous sections, we examined the synchronization between HPEs and the singular AR time series. To elaborate further on the concept of synchronized HPEs, we assess how precipitation at different locations is organized during AR-days. For that, we select the days with active ARs of level AR3 or higher, making landfall north of $47.5°$N and the respective subsequent 3 days. We acknowledge that with this approach we can only relate HPEs close to the coastline to ARs. Based on these selected days, we run a network analysis using ES with parameters $\tau_{\min} = 0$, $\tau_{\max} = 3$. Therefore, we investigate the immediate synchronization pattern of HPEs occurring simultaneously with ARs.

In Fig. 4 we show the resulting network divergence, which is characterized by a large area of negative values on the coast overland and positive values over the eastern part of the Pacific. Note that network divergence is computed by subtracting $k_i^{out} - k_i^{in}$ (out-degree minus in-degree, see Eq. (12)), therefore, areas with positive (negative) values have more outgoing (incoming) edges and can be regarded as sources (sinks) of the network. We identify a clear source in the Pacific ocean and a sink close to the western coast of NA. We find that HPEs over the Pacific occur first and are then followed by HPEs over the western parts of NA. Additionally, there is a wave-like pattern hidden in Fig. 4 (moving further east, we observe another large area of negative network divergence, followed by positive values over the Atlantic and the east coast of NA). Since our filtering only allows for interpreting the dynamics near the western coast of NA, we can only speculate about the causes of this pattern. We suspect that there are either other climate features such as synchronized ARs serving as precipitation sources in the North (e. g. as described in Mo and Lin, 2019) or a previous land-falling AR causing a cascade of synchronized HPEs traversing eastwards.

### 3.4 Network analysis of HPEs in the aftermath of AR events

After the proof of concept in the previous section, we now extend the spatial and temporal domain. Our initial analysis showed that HPEs which occur in central and eastern Canada, are synchronized with ARs land-falling on the western coast of NA considering delays in a window between $\tau_{\min} = 3$ and $\tau_{\max} = 12$ (See Fig. 2). Now, to link HPEs that are related to ARs, we choose the following setup: we consider events that occurred from 0 to 12 days after the landfall of an AR of at least level AR3 and employ ES with $\tau_{\min} = 3$ and $\tau_{\max} = 12$. With that, we assure that we keep HPEs on the coastline and in central and eastern Canada, but only allow synchronization for temporal delays larger than 3 days. Consequently, we avoid obtaining a strong signal of synchronized events along the western coast of NA (where the main synchronization of events happens during the first 3 days after the first AR day). In other words: we examine the delayed synchronization pattern of HPEs (at least 3 days between events) occurring at any time after an intense AR makes landfall. The resulting network divergence is displayed in Fig. 5a.

We identify a region of positive network divergence along the northern part of the western coast of NA and especially a region of reduced network divergence over central and eastern Canada, where the synchronization between the AR time series and the HPEs was initially discovered. To finally verify that there is a strong connection between the North American west coast, where we find a large number of outgoing edges, and central and eastern Canada, where many edges terminate, we analyze where edges that connect to central and eastern Canada originated (see out degree and red box in Fig. 5b).

Fig. 5b highlights the grid cells where edges originate that terminate in central and eastern Canada (red box). This box has been chosen based on where we have found the synchronization between the AR time series and the HPEs time series (see Fig. 2 and Fig. 3). One main source of edges to this region is the North and Northwest of the study area. This confirms that edges emerge from the region that is marked by positive values in the network divergence (where HPEs synchronized within the first 3 days of the landfall of an AR) and terminate in the red box.

In summary, we have identified a cascade of HPEs: on the first 3 days after an AR of level AR3 or higher makes landfall, HPEs are mostly occurring close to or on the coast and synchronize in this area. In the subsequent days, moisture can be

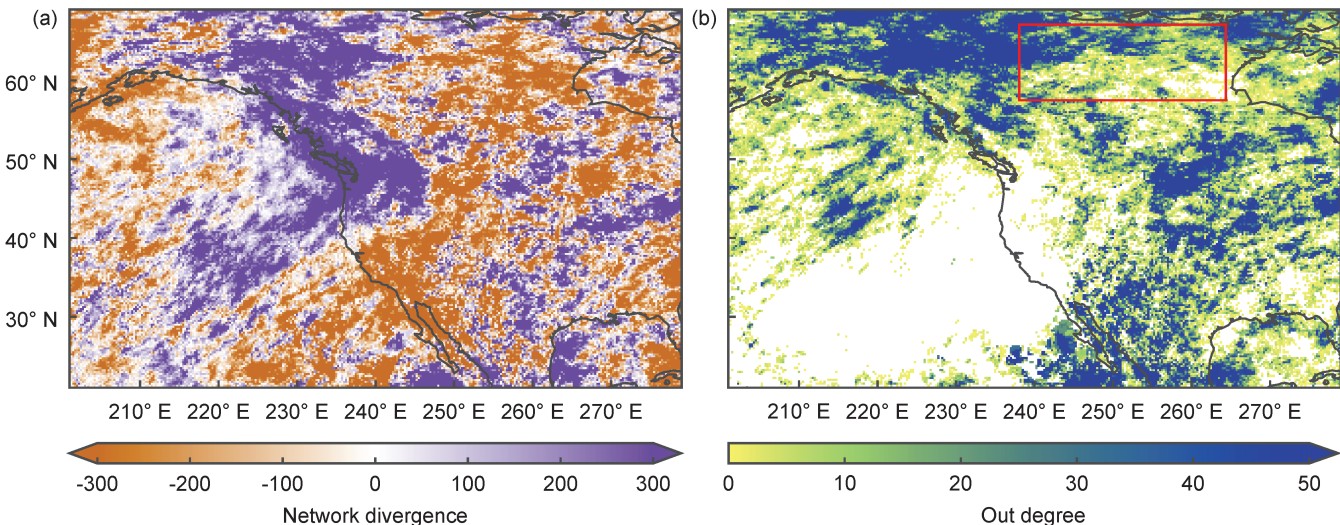

**Figure 5.** Network analysis based on event synchronization (ES) between HPEs at different locations in the aftermath of land-falling ARs. We calculate ES with $\tau_{\min} = 3$ and $\tau_{\max} = 12$, and consider all HPEs that occurred from 0 to 12 days after an AR of level AR3 or higher makes landfall north of $47.5°$N according to the SIO-R1 catalog. In the networks, only nodes with significantly directed event synchronization are connected (see Sec. 2.5 for more details). (a) Network divergence. Color bar as in Fig. 4. (b) Out degree of a directed network. Highlighted are the grid cells where edges terminating in central and eastern Canada originated (red box). This box was chosen based on the region with significant synchronization between land-falling ARs and HPEs found in Fig. 2 and Fig. 3.

transported to central and eastern Canada and cause HPEs there. This takes place between 3 and 12 days after the first AR-induced precipitation on the coastline.

### 3.5 Synoptic conditions facilitating AR-induced HPEs in central and eastern Canada

Measuring the synchronization between time series revealed the spatial extent as well as the temporal dimensions of heavy precipitation related to land-falling ARs over NA. A delayed synchronization pattern between ARs making landfall on the western coast of NA and HPEs in central and eastern Canada was identified. To examine the climatic drivers leading to this long-ranged correlation, we study the synoptic conditions of different climatological variables for three types of events, defined as follows,

1. type I event: an AR makes landfall on the western coast of NA and synchronizes with HPEs in central and eastern Canada in the subsequent 3 to 12 days.

2. type II event: an AR makes landfall on the western coast of NA but does not synchronize with HPEs in central and eastern Canada in the subsequent 3 to 12 days.

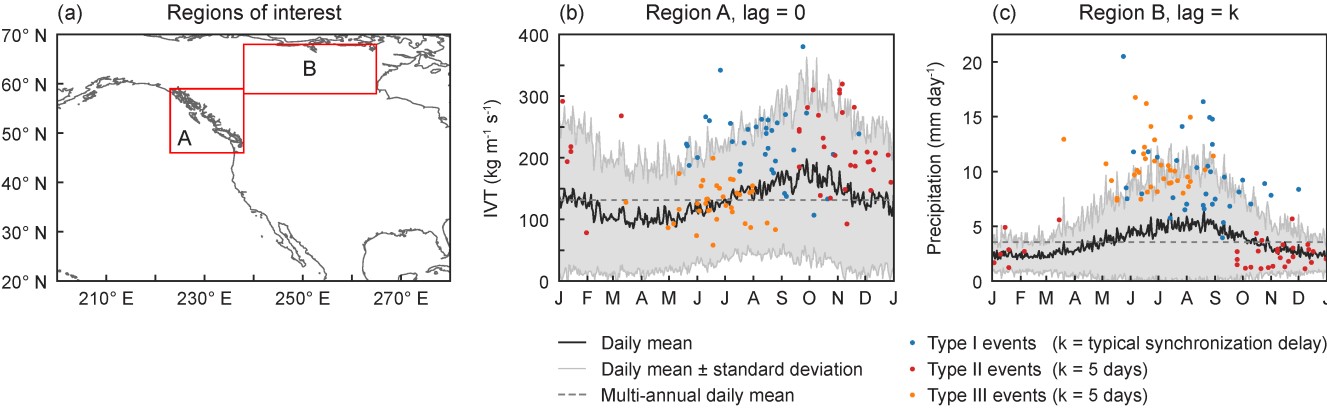

**Figure 6.** (a) Regions of interest for the study of the synoptic conditions during and after the landfall of ARs. Region $A$ ($46°$N to $59°$N latitude and $223°$E to $238°$E longitude) delimits the area where ARs (with land-falling latitude $\geq 47.5°$) make landfall. Region $B$ ($58°$N to $68°$N latitude and $238°$E to $265°$E longitude) delimits the area where the delayed synchronization between ARs and HPEs occurs. (b) Annual cycle at daily resolution of the mean IVT in region $A$ (solid dark line). The shading encloses one standard deviation from the daily mean and the dashed line indicates the multi-annual mean IVT of the region. The dots show the IVT values for days with ARs that synchronize with HPEs (in blue), days with ARs that do not synchronize with HPEs (in red), and days preceding HPEs without land-falling ARs (in orange) (c) Annual cycle at daily resolution of the mean precipitation in region $B$, with the same conventions as in panel (b). Note that the dots in panel (c) are lagged with respect to the dots in panel (b) by a value of $k \in \mathbb{N}$ that depends on the type of event, such that the dots in panel (b) show the IVT conditions in the western coast of NA and the dots in panel (c) show the subsequent precipitation conditions in central and eastern Canada. For more details on the types of events and the corresponding value of $k$, see Secs. 2.3 and 3.5.

 3. type III event: HPEs occur in central and eastern Canada but no AR made landfall on the western coast of NA during the previous 12 days.

To carefully choose the time points corresponding to each type of event, we first identify specific times with high event synchronization between land-falling ARs and HPEs in central and eastern Canada. We do so by using Eq. (5) with $l = 3$, $\rho = 0.9$, $\tau_{\min} = 3$, $\tau_{\max} = 12$ (to match the spatial pattern found in Figs. 2 and 3), and the region of interest as the box in Fig. 5b, which we denote as region $B$ (see Fig. 6a). Note that the resulting $\{ES^{0.9}_{\mu \to B}\}_{\mu=1,...,n_l}$ is a sequence that gives the total number of HPEs in region $B$ that were preceded and uniquely associated with the AR event $\mu$ at a significance level $\alpha \leq 0.1$. We identify ARs whose total number of associated HPEs is above the $80$th percentile of the nonzero values of this sequence and we get 35 AR events higly synchronized with HPEs in central and eastern Canada. The land-falling times of these ARs are the time points of type I events. We also use Eq. (6) to determine the typical synchronization delay between region $B$ and each of these highly synchronized ARs, with the most common value being 5 days. We then select the 35 most intense AR events for which $ES^{0.9}_{\mu \to R} = 0$, i.e. those that did not synchronize with HPEs in central and eastern Canada, and we select their land-falling times as the time points of type II events. Finally, we identify the time points of type III events as the 35 days

with the highest number of HPEs in central and eastern Canada that occurred in the absence of land-falling ARs during a time window of 12 precedent days.

We use these 3 types of events to analyze the antecedent IVT over the western coast of NA and the subsequent precipitation over central and eastern Canada, and thus define the two regions of interest shown in Fig. 6a. Region $A$, where we analyze IVT anomalies, covers the area where the ARs make landfall north of $47.5°$N (from $46°$N to $59°$N latitude and from $223°$E to $238°$E longitude). Region $B$, where we study precipitation anomalies, delimits the area where the delayed synchronization pattern between ARs and HPEs was identified (from $58°$N to $68°$N latitude and from $238°$E to $265°$E longitude). In Fig 6b we show the annual cycle at daily resolution of the mean IVT in region $A$ (solid dark line). The shading encloses one standard deviation from the daily mean and the dashed line indicates the multi-annual mean IVT of the region. The dots show the IVT values of the 3 types of events previously described. Similarly, Fig 6c shows the annual cycle at daily resolution of the mean precipitation in region $B$ and the precipitation values of the 3 types of events. It is important to clarify that the dots in panel (c) are time-delayed with respect to the dots in panel (b), as follows: *i)* For type I events, the dots in panel (b) are shown on the day the AR makes landfall and the dots in panel (c) are shown after the respective typical synchronization delay ($k \in \{3, 4, \ldots, 12\}$). *ii)* For type II events, the dots in panel (b) are also shown on the day the AR makes landfall but the corresponding dots in panel (c) are shown 5 days later ($k = 5$). *iii)* For type III events, the dots in panel (b) are shown 5 days before the occurrence of HPEs in central and eastern Canada and the corresponding dots in panel (c) are shown on the day when the HPEs were recorded ($k = 5$).

First, note the clear imprint of ARs on IVT values for type I and II events (Fig 6b, blue and red dots). Whether or not the land-falling AR synchronizes with HPEs in central and eastern Canada, the IVT values in region $A$ are above the daily and the multi-annual mean with very few exceptions. Highly synchronized ARs, corresponding to type I events, have particularly anomalous values of IVT, in most cases exceeding the mean climatology by one standard deviation or more. On the contrary, the IVT values of most type III events (orange dots) are around or below the daily and multi-annual mean IVT of the region, confirming the absence of a precedent AR making landfall on the coast for this type of events. The delayed precipitation over central and eastern Canada associated to each type of events is displayed in Fig. 6. For type I events, as expected, the precipitation values are above the daily and the multi-annual mean, with anomalies that frequently exceed the mean climatology by more than one standard deviation. Conversely, type II events have average or lower precipitation values which is consistent with the absence of a synchronization pattern between land-falling ARs and HPEs. Not surprisingly, type III events referring to HPEs with no precedent ARs also have high precipitation anomalies.

Besides this proof of concept on the effectiveness of our methodology to identify types of events and synchronization delays, Fig. 6 also reveals a key characteristic of the ARs that are highly synchronized with HPEs in central and eastern Canada: their seasonality. These ARs, which define the type I events, occur most likely during July, August, and September, and can be characterized as intense, long-lasting, late-summer ARs. Type II events that do not contribute to the observed synchronization pattern in central and eastern Canada, are caused by ARs with similar strength and persistence but making landfall during the early winter, i.e. in October, November, and December. Lastly, type III events describing HPEs that are not preceded by ARs are more common during the early summer months of June, July, and August.

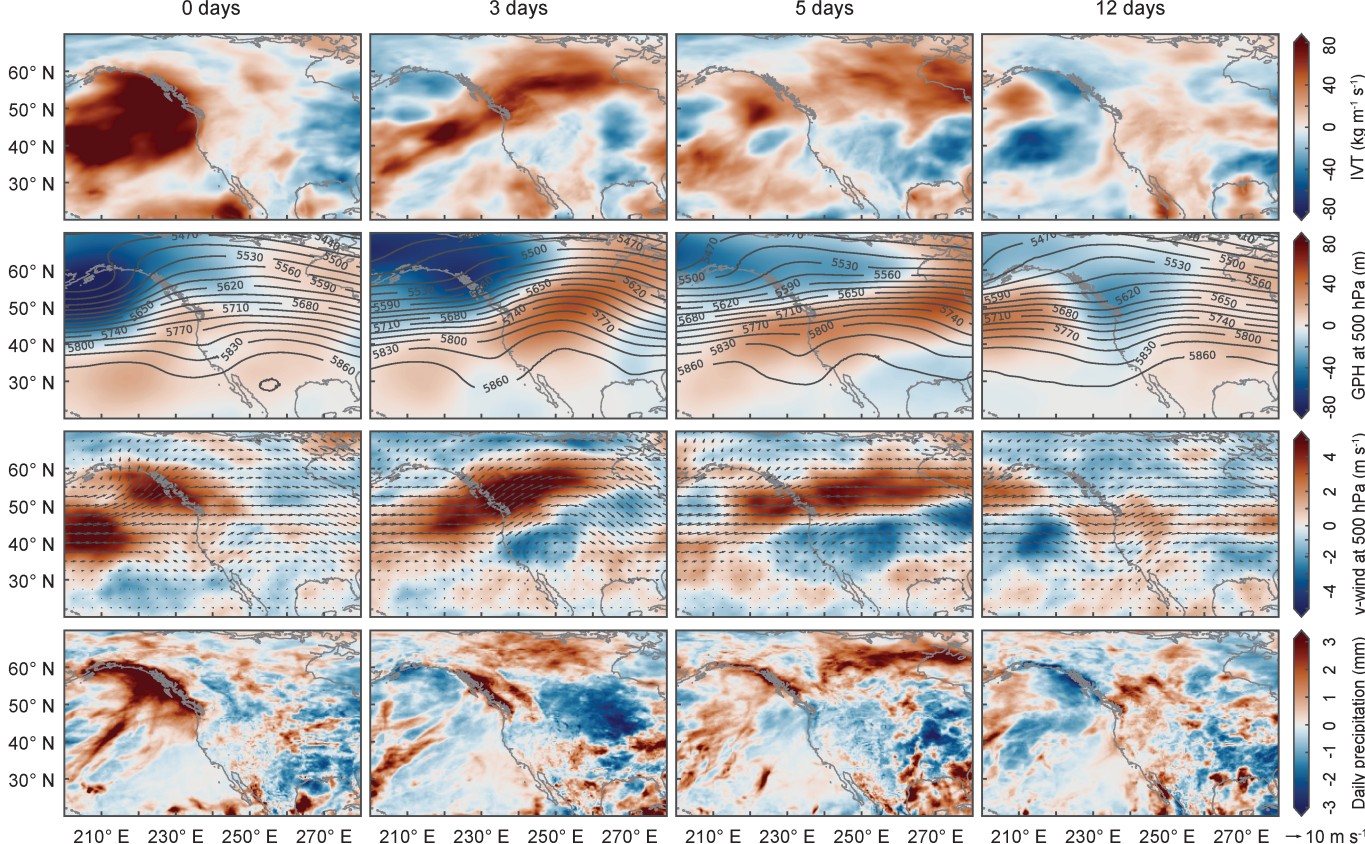

**Figure 7.** IVT, geopotential height at 500 hPa, wind at 500 hPa, and precipitation anomalies (from top to bottom), from 0, 3, 5, and 12 days (from left to right) after the landfall of ARs leading to a delayed synchronization pattern with HPEs in central and eastern Canada. Only ARs of level AR3 or higher with land-falling latitude north of $47.5°$N are considered. In the second row, the shading indicates the anomaly of the geopotential height at 500 hPa and the contours show the mean geopotential height. In the third row, the shading indicates the anomaly of the meridional wind at 500 hPa and the arrows show the mean wind field.

To reveal the synoptic conditions facilitating the delayed effect of ARs in the precipitation over central and eastern Canada, we compute composite anomalies of IVT, geopotential height at 500 hPa, wind at 500 hPa, and precipitation, on the days of type I events (when the highly synchronized ARs made landfall) and for the following 3, 5, and 12 days. The results are shown in Fig. 7. Similar figures for type II and III events can be found in the supplementary material (see appendix A, Figs. A4 and A5).

In Fig. 7, we first present the temporal evolution of the IVT anomalies (top row) after the landfall of highly synchronized ARs on the western coast of NA. The high positive IVT anomaly on the Pacific when lag = 0 days is a clear characteristic of land-falling ARs. Noteworthy is that, in the following days, this anomalous water vapor influx is able to penetrate the continent through the topographic gap previously identified by Rutz et al. (2014) and traverses the mainland reaching central and eastern

Canada, where the synchronization pattern was found. In the second row of Fig. 7 the geopotential height at 500 hPa is assessed for varying delays as for the IVT. The trough together with the strong negative anomalies in the northwest of the study region at the moment of land-fall indicates the position of the cold front driving the ARs. In the following days, the cold front digs into Canada as a ridge builds in central and eastern United States. This configuration of the geopotential height field during highly synchronized ARs implies a mid-level pressure dipole that traverses the continent accompanied by a southwesterly steering wind (third row of Fig. 7), bringing warm moist air from the western coast into the northern regions of NA. Under this circumstances, high precipitation anomalies occur just downwind of the trough, first at the coast and then over region $B$, where we identified the delayed synchronization pattern between ARs and HPEs (fourth row of Fig. 7). These synoptic conditions, that are exclusive to highly synchronized ARs (see Figs. A4 and A5) and have already been identified as conducive to the occurrence of seasonal precipitation extremes over Canada (Tan et al., 2019), explain the physical mechanism by which land-falling ARs serve as moisture sources of HPEs in the northern regions of NA.

## 4   Conclusions

In this study, we have investigated the influence of ARs on the large-scale spatio-temporal synchronization patterns of HPEs over NA. For this purpose, we have first analyzed if there is a significant association between ARs making landfall on the western coast of NA and HPEs over the coastal and continental regions. Employing event synchronization (ES), we have revealed timescale-dependent spatial patterns of HPEs that are significantly correlated with ARs making landfall north of $47.5°$N: i) immediately after an AR makes landfall on the coast, HPEs synchronize over the coastal areas. ii) Then, from 3 days after the landfall, the synchronization close to the coast decreases significantly and only HPEs associated with more persistent ARs remain. iii) From 3 to 12 days after an AR makes landfall, a synchronization pattern between land-falling ARs and HPEs in central and eastern Canada emerges. These results have been reproduced using an alternative AR catalog, establishing the robustness of our findings.

After examining the synchronization of HPEs with the time series of land-falling ARs, we have analyzed the organization of HPEs on the day of landfall and the subsequent days. For this, we have evaluated directed ES between time series of HPEs at different locations and for different temporal lags after the landfall of ARs. Based on that, we have first constructed a complex network considering a time window from 0 to 3 days after landfall. This result confirmed the common knowledge of the relation between ARs and HPEs in western NA. Initially, HPEs occurring simultaneously with ARs are synchronized over the eastern Pacific Ocean. They are then followed by synchronized HPEs on the western coast of NA. By examining a second complex network based on HPEs occurring at any time after the landfall of an intense AR, but only allowing synchronization with a delay from 3 to 12 days, we have uncovered a strong connection between HPEs on the North American west coast and in central and eastern Canada: moisture from ARs land-falling along the coast can be transported to central and eastern Canada and cause HPEs there.

To further investigate this result, we identified specific days with high event synchronization between land-falling ARs of level AR3 or higher and HPEs in central and eastern Canada. Then, we used these time points to analyze the composite anoma-

lies of vertically integrated water vapor transport (IVT), geopotential height at 500 hPa, wind at 500 hPa, and precipitation, on the day of landfall and during the subsequent 3, 5 and 12 days. Our approach yielded two key findings regarding the climatic conditions that facilitate AR-induced HPEs in central and eastern Canada: *i)* intense, long-lasting, late-summer ARs making landfall north of 47.5°N on the western coast of NA are the ones leading to the occurrence of delayed HPEs in central and eastern Canada, and *ii)* such ARs are driven by a cold front digging from the Northeast Pacific Ocean into Canada as a high-pressure region builds in central and eastern United States. This mid-level pressure dipole traverses the continent accompanied by a southwesterly steering wind, bringing the warm moist air deposited on the coast by the land-falling ARs into central and eastern Canada, and facilitating synchronized but delayed HPEs there. These particular synoptic conditions, that are consistent with the seasonality of the identified ARs, explain the physical mechanism by which late-summer ARs serve as moisture sources of HPEs in the northern regions of NA. However, whether or not these ARs remain as identifiable objects following landfall remains an open question that requires a different catalog than SIO-R1 to be addressed. More specifically, a catalog that tracks the ARs not only until they make landfall, but also as they penetrate the continent.

In summary, we have studied the spatio-temporal synchronization pattern of HPEs induced by ARs, revealing its extent and its temporal evolution. We have shown that the impact of ARs making landfall on the western coast of NA is not limited to these areas, since they can be accompanied by delayed but significantly synchronized HPEs in the continental regions. In particular, we have identified a cascade of synchronized HPEs: on the first 3 days after an AR makes landfall, HPEs occur and synchronize along the coast. In the subsequent days, this moisture can be transported to central and eastern Canada and cause synchronized HPEs there. Our results illustrate the role of ARs in the distribution of HPEs over NA, not only on the west coast but also over the continental regions through inland penetration of IVT. The findings presented in this work should be considered to better anticipate the evolution of the climate dynamics of the region and the associated impacts in the precipitation patterns in the context of a warming atmosphere, for which we expect an increased frequency and strength of the ARs as well as a northward shift of the locations where the ARs make landfall.

*Code and data availability.* The code is available from the authors upon request. The analysis was conducted with Python and supported by the Python package Pyunicorn Donges et al. (2015). All data sets are publicly available. The ERA5 reanalysis data sets can be downloaded at https://cds.climate.copernicus.eu/. The SIO R1 Catalog can be accessed via https://weclima.ucsd.edu/data-products/.

*Author contributions.* SV and FW conducted the analysis and wrote the manuscript. SV prepared the figures with the help of FW. DT prepared the section and table for the IPART catalog. All authors reviewed and improved the manuscript. SV conducted the reviews.

*Competing interests.* The authors declare that they have no competing interests.

*Acknowledgements.* This research has been funded by the BMBF grant climXtreme (No. 01LP1902J) "Spatial synchronization patterns of HPEs" and by DFG research training group GRK 2043/1 "Natural risk in a changing world (NatRiskChange)". NB acknowledges funding from the Volkswagen Foundation. SV acknowledges and appreciates the stimulating and fruitful discussions with Dr. Tobias Braun.

# Appendix A

## A1   Analysis of the impact of latitudinally categorized ARs.

In the main manuscript, we have only used ARs making landfall north of $47.5°$. We based that on the finding that the number of grid cells at which HPEs are significantly correlated with ARs is not increased by including ARs making landfall at lower latitudes. For that, we have step-wise included more ARs (with a $2.5°$ step size) and counted the number of significant grid cells in central and eastern Canada (for the spatial extent, see red box in Fig. 5b). As an illustration of how the results appear when including all ARs, we have run the analysis evaluating the synchronization between HPEs and ARs making landfall anywhere on the western coast of NA. We show Fig. A1, featuring otherwise the same key findings as in Fig. 2.

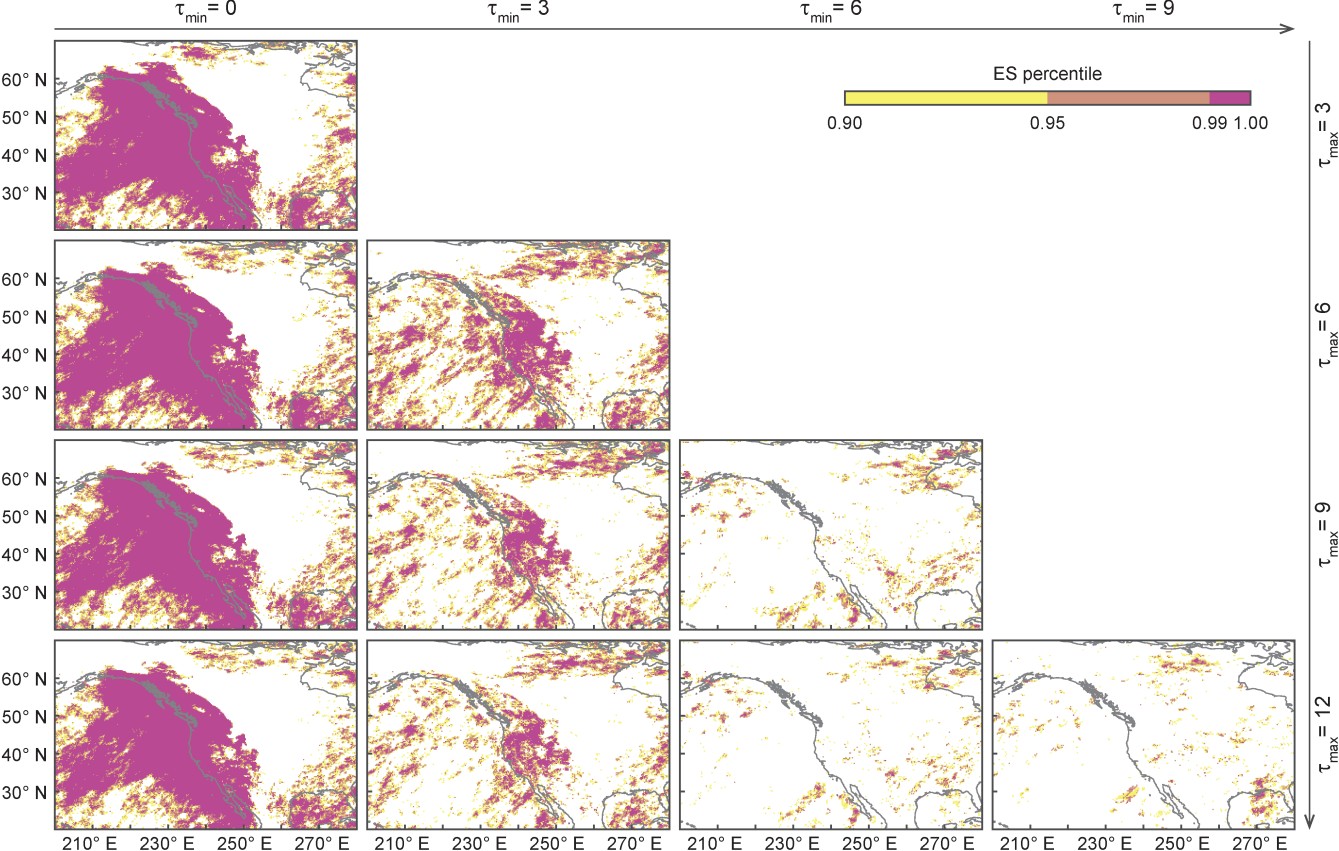

**Figure A1.** Event synchronization (ES) between ARs making landfall on the western coast of NA and HPEs. We use the SIO-R1 catalog of land-falling ARs and consider all ARs of level AR3 or higher. Different values of $\tau_{min}$ and $\tau_{max}$ are considered to calculate ES in each panel: $\tau_{min}$ increases from left to right and $\tau_{max}$ from top to bottom. Color bar as in Fig. 2.

As the results shown in Fig. 2 and Fig. A1 are not visually distinguishable in central and eastern Canada and the number of grid cells exhibiting significant synchronization does not increase in that region, we assume our choice is robust and proceeded with the subset of ARs for the main analysis.

## A2 Dependence on the choice of the AR catalog

As mentioned in the introduction, a plethora of work has analyzed how the choice of an AR detection algorithm affects the outcome of an analysis (Shields et al., 2018; Rutz et al., 2019; O'Brien et al., 2022). For this reason, we have re-run the whole analysis for a systematically different AR catalog (for details see Sec. 2). Whereas in the main manuscript we utilized the SIO-R1 catalog by Gershunov et al. (2017), here we feature the analysis carried out with a catalog based on the IPART algorithm (Xu et al., 2020; Traxl, 2022).

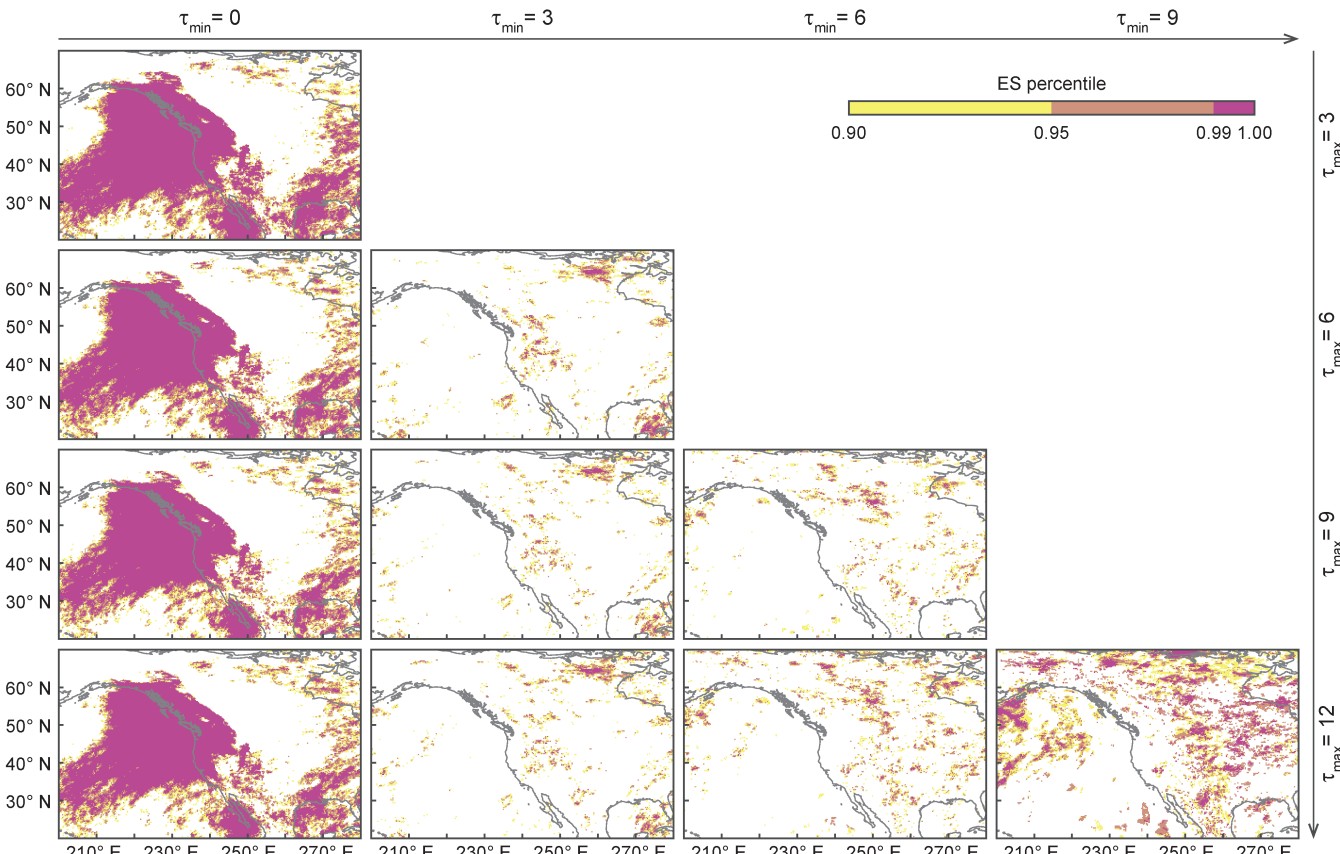

**Figure A2.** Event synchronization (ES) between ARs making landfall on the western coast of NA and HPEs. We use the IPART catalog of land-falling ARs and consider all ARs of level AR2 or higher. Different values of $\tau_{min}$ and $\tau_{max}$ were considered to calculate ES in each panel: $\tau_{min}$ increases from left to right and $\tau_{max}$ from top to bottom. Color bar as in Fig. 2.

To verify that we find, in principle, the same results using this alternative approach, we again show the results of assessing the synchronization between land-falling ARs and HPEs, and as in the previous section, we consider all ARs making landfall anywhere on the western coast of NA. The results are shown in Fig. A2 and, as expected, are visibly different to some extent, so we must acknowledge that we had to adapt the parameters of the analysis. In particular, for the IPART catalog, we reduced the considered lower threshold for the AR level. Therefore, Fig. A2 is based on ARs of level AR2 and higher, whereas Fig. A1, and Fig. 2 are based on ARs of level AR3 and higher. We assume that this is due to two reasons: first, ARs identified by the IPART catalog have, on average, a shorter persistence in comparison to the ones listed in the SIO-R1 catalog. Therefore, ARs are often ranked higher in the SIO-R1 catalog (persistence is one criterion on the AR scale by Ralph et al., 2019). Second, the IPART algorithm identified significantly fewer ARs, which leads to a more sparse AR time series. Then, filtering out many ARs may decrease the ES score due to the sparsity of the time series.

Aside from this adaptation, which we consider reasonable, we again find a region of synchronization between land-falling ARs and HPEs in central and eastern Canada. Note that the results show striking qualitative similarity (the signal is strongest for $\tau_{\min} \geq 3$ and $\tau_{\max} = 12$), and for $\tau_{\min} \geq 3$ the signal close to the coast vanishes/gets less significant). Therefore, we consider the results based on the IPART catalog (Fig. A2) comparable to the results featured in the main manuscript (Fig. 2).

To be transparent regarding the construction of the IPART catalog we refer to Xu et al. (2020) and the chosen parameters below, which are mostly the default parameters of the algorithm.

## A3  Dependence on the choice of the AR level

In the main manuscript we have stated that the synchronization pattern observed in the western coast of NA between land-falling ARs and HPEs occurring between $0$ and $3$ after the landfall does not change if ARs of the lower levels AR1 and AR2 are excluded from the calculations. This statement is based on the results shown in Fig. A3, which displays the grid points whose time series of HPEs are significantly synchronized with the AR time series, when $\tau_{\min} = 0$ and $\tau_{\max} = 3$. In panel (a) all the ARs are considered, and for the subsequent panels, ARs of the lower levels are step-wise excluded such that the lower limit of the considered AR level increases. Note that the most prominent synchronization pattern is always present along the western coast of NA, which is expected given the direct impact of ARs in the inmediate HPEs of this area (Neiman et al., 2008; Gershunov et al., 2017; Waliser and Guan, 2017; Ralph et al., 2019). However, excluding ARs of the lower levels AR1 and AR2 does not change this result, as only considering ARs from levels AR4 and AR5 does. Including ARs of the lower levels for the calculations of ES introduces noise into the results, especially in eastern NA, where the synchronization pattern is not related to ARs making landfall on the West Coast but rather to ARs and extra-tropical cyclones making landfall on the East Coast. On the other hand, only considering ARs of leves AR4 and AR5 reduces the number of events and makes the AR time series too sparse to retain the statistical significance of the results. Since panel (c) exhibits an intermediate pattern between these 2 scenarios, we have selected ARs of level AR3 or higher for our analysis.

**Table A1.** Parameters used to create the IPART AR catalog.

| Parameter name | Short description of the parameter | Parameter value | Unit | Step of the IPART algorithm |
|---|---|---|---|---|
| kernel | List of integers specifying the shape of the structuring element used in the gray erosion process | [16, 6, 6] | - | Top-hat by Reconstruction computation on IVT data |
| shift_lon | Shifts data along longitude dimension | 80 | degrees longitude | Top-hat by Reconstruction computation on IVT data |
| thres_low | Define AR candidates as regions >= this anomalous IVT | 1 | $kg\,m^{-1}\,s^{-1}$ | Detect AR appearances from Top-hat by Reconstruction output |
| min_area | Drop AR candidates smaller than this area | 500.000 | $km^2$ | Detect AR appearances from Top-hat by Reconstruction output |
| max_area | Drop AR candidates larger than this area | 18.000.000 | $km^2$ | Detect AR appearances from Top-hat by Reconstruction output |
| min_LW | Minimal length to width ratio | 2 | - | Detect AR appearances from Top-hat by Reconstruction output |
| min_lat | Exclude ARs whose centroids are lower than this latitude | 20 | degrees latitude | Detect AR appearances from Top-hat by Reconstruction output |
| max_lat | Exclude ARs whose centroids are higher than this latitude | 80 | degrees latitude | Detect AR appearances from Top-hat by Reconstruction output |
| min_length | ARs shorter than this length are treated as relaxed | 2,000 | km | Detect AR appearances from Top-hat by Reconstruction output |
| min_length_hard | ARs shorter than this length are discarded | 1,500 | km | Detect AR appearances from Top-hat by Reconstruction output |
| rdp_thres | Error when simplifying axis using rdp algorithm | 2 | degrees latitude/longitude | Detect AR appearances from Top-hat by Reconstruction output |
| fill_radius | Number of grids as radius to fill small holes in AR contour | 4 | - | Detect AR appearances from Top-hat by Reconstruction output |
| single_dome | Do peak partition or not, used to separate systems that are merged together with an outer contour | False | - | Detect AR appearances from Top-hat by Reconstruction output |
| edge_eps | Minimal proportion of flux component in a direction to total flux to allow edge building in that direction | 0.4 | - | Detect AR appearances from Top-hat by Reconstruction output |
| zonal_cyclic | Whether to treat the data as zonally cyclic | True | - | Detect AR appearances from Top-hat by Reconstruction output |
| TIME_GAP_ALLOW | Gap allowed to link 2 ARs | 6 | hours | Track ARs at individual time steps to form tracks |
| TRACK_SCHEME | Tracking scheme | simple | - | Track ARs at individual time steps to form tracks |
| MAX_DIST_ALLOW | Maximal Hausdorff distance to define a neighborhood relationship | 1200 | km | Track ARs at individual time steps to form tracks |
| MIN_DURATION | Minimal duration to keep a track | 0 | hours | Track ARs at individual time steps to form tracks |
| MIN_NONRELAX | Minimal number of non-relaxed records in a track to keep a track. | 0 | - | Track ARs at individual time steps to form tracks |

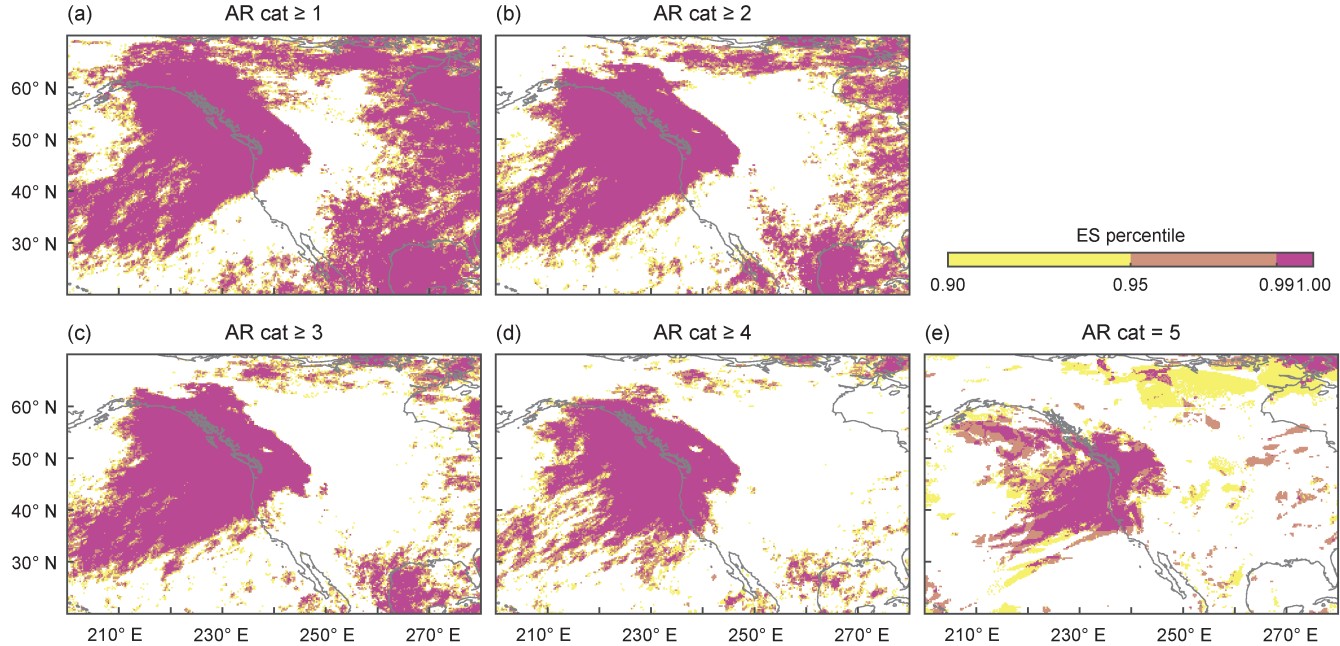

**Figure A3.** Event synchronization (ES) between ARs making landfall on the western coast of NA and HPEs. We use the SIO-R1 catalog of land-falling ARs but only consider ARs making landfall north of $47.5°$N. ES is calculated with $\tau_{min} = 0$ and $\tau_{max} = 3$. From (a) to (e) the lower limit of the considered AR level increases: (a) ARs of level AR1 and higher e.g. all ARs, (b) ARs of level AR2 and higher, rest accordingly. Color bar as in Fig. 2.

## A4 Synoptic conditions during type II and III events

In the main manuscript we revealed the synoptic conditions facilitating the delayed effect of ARs in the precipitation over central and eastern Canada. We did so by analyzing composite anomalies of IVT, geopotential height at 500 hPa, wind at 500 hPa, and precipitation, on the days of type I events (when the highly synchronized ARs made landfall) and for the following 3, 5, and 12 days. Now, we contrast those results with the composite analysis for type II and III events, i.e. for days after the landfall of ARs that did not synchronize with HPEs in central and eastern Canada, and for days before HPEs in central and eastern Canada that occurred in the absence of a land-falling AR (see Sec. 3.5 for a detailed description of the types of events).

In Fig. A4, we first present temporal evolution of the synoptic conditions after the landfall of ARs that did not synchronize with HPEs in central and eastern Canada. As in Fig. 7, when lag = 0 days, there is a high positive IVT anomaly on the Northeastern Pacific accompanied by a cold front and a southwesterly steering wind driving the ARs to the western coast of NA. Moreover, on the day of landfall, the mid-level pressure dipole identified in Fig. 7 is present and determines the location of HPEs along the western coast of NA. However, the striking difference between ARs leading to type II events and those leading to type I events is the evolution of this pressure dipole in the days following landfall. Note that the cold front vanishes instead

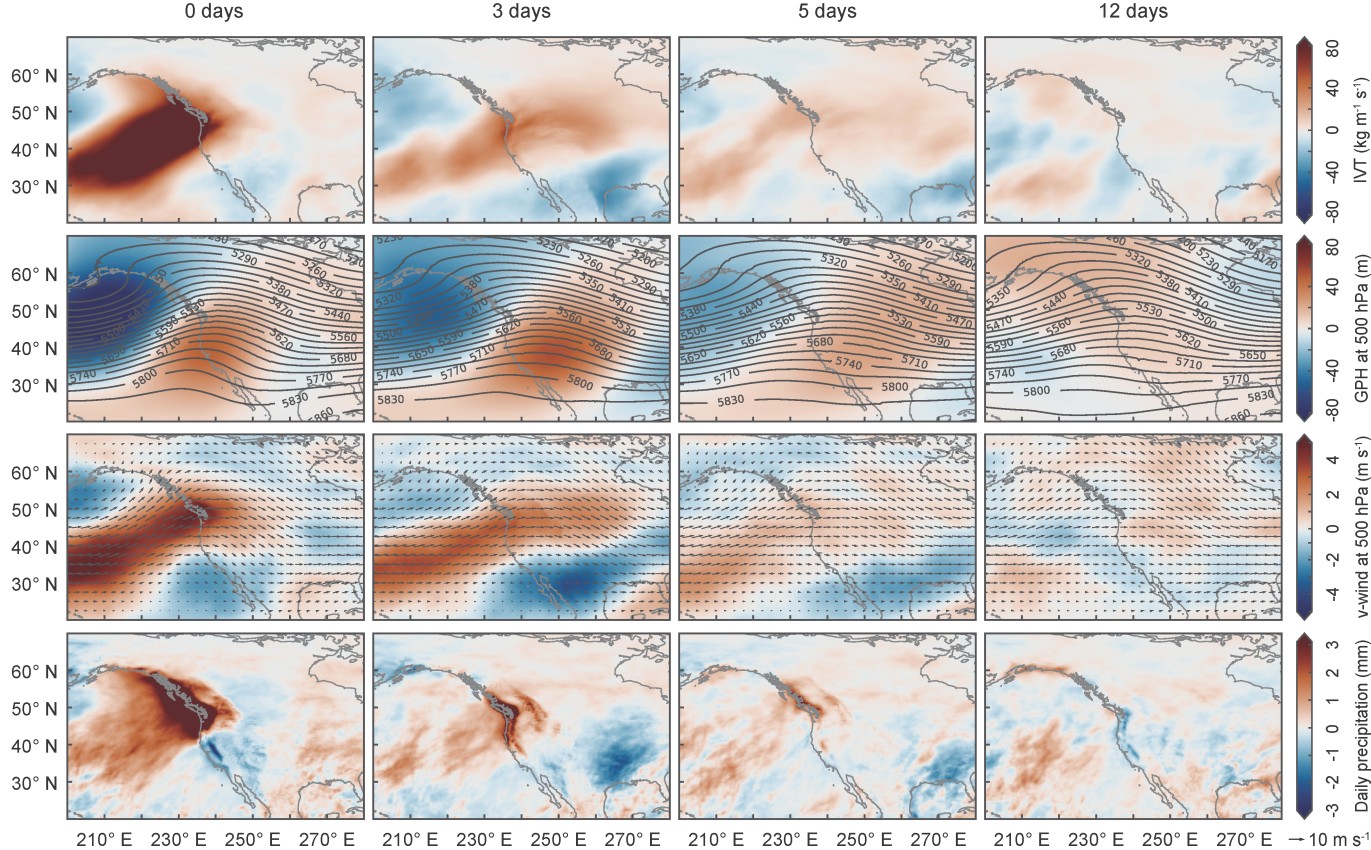

**Figure A4.** Same as in Fig. 7 but for land-falling ARs that do not synchronize with HPEs in central and eastern Canada.

of digging into Canada, preventing the anomalous influx of water vapor from reaching the northern parts of North America, and therefore resulting in the absence of HPEs in central and eastern Canada.

Secondly, in Fig. A5 we present the temporal evolution of the synoptic conditions preceding HPEs in central and eastern Canada that occurred in the absence of land-falling ARs. Note the negative IVT, wind, and precipitation anomalies that are always present over the Northeastern Pacific and the western coast of NA, which are attributable to the warm front located in the northwest of the scene. These specific synoptic conditions are a clear indicator of the absence of land-falling ARs during the 12 precedent days considered for the analysis. For type III events, the climatological drivers are more likely related to summertime convective processes forced by the trough located over central Canada when the HPEs occur (right column) (Raddatz and Hanesiak, 2008).

## A5 Precipitation anomalies for increasing AR levels

As stated in the main manuscript, only considering ARs of level AR4 and higher, or even examining only ARs of level AR5, does not lead to significant correlation between land-falling ARs and HPEs. We suspect that first, the sparsity of the time series

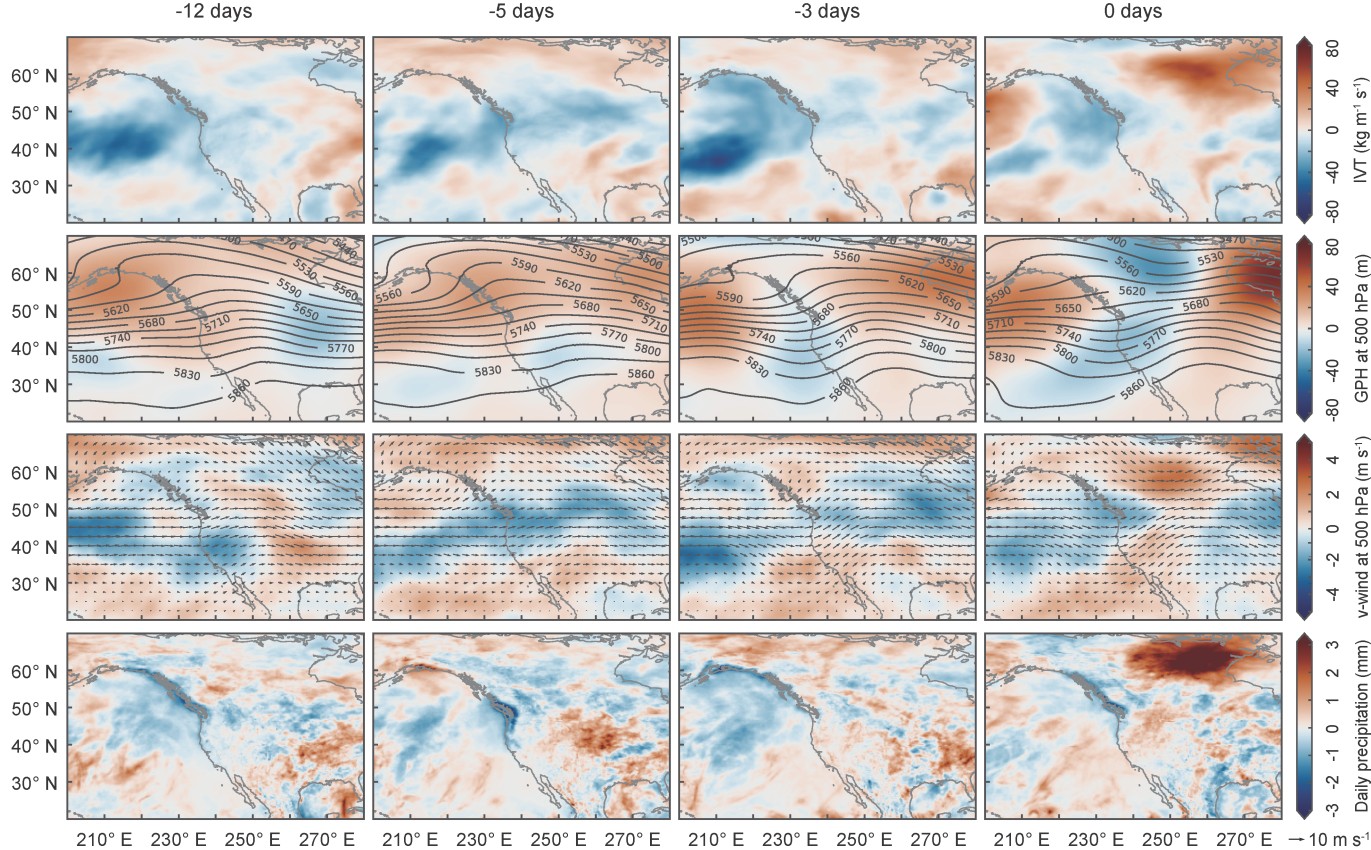

**Figure A5.** IVT, geopotential height at 500 hPa, wind at 500 hPa, and precipitation anomalies (from top to bottom), from 12, 5, 3, and 0 days (from left to right) before the occurrence of HPEs in central and eastern Canada with no precedent land-falling ARs on the western coast of NA at locations north of 47.5°N, according to the SIO-R1 catalog. Shading, contours and arrows as in Fig. 7

does not allow for significant ES scores but also that HPEs do not *only* occur after such strong ARs. To give an argument that these ARs still contribute to the HPEs, we studied the precipitation anomalies in the aftermath of just these exceptional AR
events. The results are shown in Fig. A6.

Here, as in Fig. 7, the delay grows from left to right and the intensity increases from top to bottom. As there are only a few events, the pattern differs visibly between the different parameter settings, but nearly all configurations show a considerable signal of above-average precipitation in central and eastern Canada. Therefore, we conclude that these strong AR events are one integral part of the identified precipitation scheme but explain only one puzzle piece and, thus, do not account just by
themselves for the increased precipitation in central and eastern Canada.

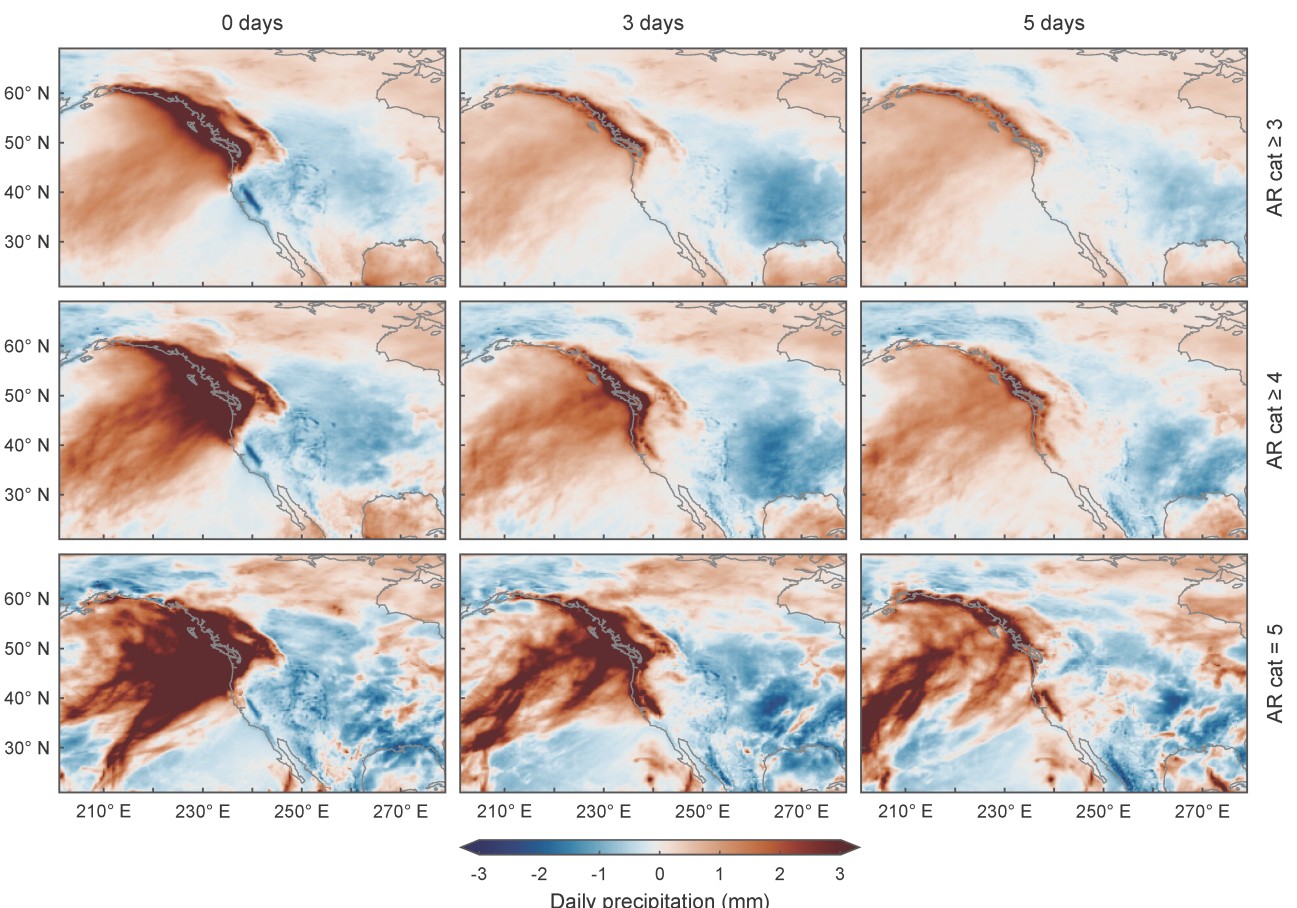

**Figure A6.** Precipitation anomalies, from 0 (left column), 3 (middle column), and 5 (right column) days after the first day of landfall of an AR of level AR3 or higher (top row), level AR4 or higher (middle row), and level AR5 (bottom row). We only consider ARs land-falling at locations north of $47.5°$N according to the SIO-R1 catalog.

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
