# Peer review of "The role of atmospheric rivers in the distribution of heavy precipitation events over North America"

_EGUsphere, 2022_

## Author Response (AR1)

**THE ROLE OF ATMOSPHERIC RIVERS IN THE DISTRIBUTION OF HEAVY PRECIPITATION EVENTS OVER NORTH AMERICA**

**Response to reviewers' comments**

Sara M. Vallejo-Bernal

March 2023

This document contains responses to the review of the manuscript "Spatio-temporal synchronization of heavy rainfall events triggered by atmospheric rivers in North America". It is an attachment to the revised version of the manuscript submitted to the Hydrology and Earth System Sciences Journal.

Every single one of the referees' comments were addressed and are listed below along their corresponding responses. Following the recommendations of both reviewers, mayor changes that improved the results and added value to them within the context of the existing literature were done. I would like to sincerely thank the reviewers for the work and effort put into evaluating this manuscript.

**Reviewer 1.**

**General comments:**

This manuscript identifies a synchronization between landfalling atmospheric rivers (ARs) on the west coast of North America and heavy precipitation in central Canada. The authors use a novel technique of Event Synchronization to assess the timing of heavy precipitation across North America to a time series of landfalling ARs. Due to the relative timing of these events the authors conclude that there is a linkage between these events, suggesting a moisture pathway following AR landfall. The authors utilize network analysis to examine the statistical relationship in the timing of heavy rainfall and landfalling ARs, identifying the sequence of events: central Canadian precipitation is preceded by west coast precipitation when associated with high ranked ARs. Composite analysis is undertaken to present the synoptic scale conditions during and following landfall of high ranking ARs.

**Specific comments:**

Overall, this manuscript is presented very well, in a clear, precise manner. The writing style is generally excellent with minor grammatical errors identified below. Figures are presented well and the appendices complement the work well with important caveats addressed such as reproducibility with a second AR detection method. The methodology of event synchronization is certainly novel and its application in the analysis of precipitation time series is highlighted here. The applicability of the final conclusions, however, appear to overstate the scientific discoveries made throughout the manuscript. While the main conclusion of precipitation synchronization and AR landfall is very interesting, it raises many

questions and does not represent a significant contribution by itself. The composite analysis contains confusions/errors (noted below) which does not allow for a suitable discussion of the related atmospheric dynamics. A major revision of the composite analysis is required for this paper to provide a substantial scientific contribution; to accurately describe the synoptic conditions during high ranking ARs that facilitate inland penetration of moisture following landfall.

It would also be beneficial for the authors to examine and discuss some of the literature on inland penetrating ARs and AR lifecycles in North America (such as Rutz et al., 2015; https://doi.org/10.1175/MWR-D-14-00288.1). This study has the potential of being beneficial to the AR community if it is presented within the context of modern AR studies. Some of the questions that arise are:

- Do the ARs remain as identifiable objects following landfall and during the central Canadian precipitation or have they undergone termination? Post-termination precipitation is an interesting concept that the presented findings may suggest.
- What are the synoptic conditions that allow for a high ranked AR to cause precipitation in central Canada? Are there some ARs that do not cause this precipitation, what atmospheric conditions allow for this to occur?

R / First, we want to thank the reviewer for this extensive and comprehensive review of our study, which has helped us to significantly improve our methods, results and discussion.

We agree with the main concerns of the reviewer:

In the first version of this manuscript, we calculated event synchronization between landfalling ARs and heavy precipitation over North America. However, during the calculation process, we did not identify the ARs leading to the observed synchronization patterns. Consequently, for the composite analysis, we used all ARs of level AR3 or higher, even though most of them were not synchronized with heavy precipitation over Canada, resulting in an inaccurate representation of the atmospheric dynamics. We thank the reviewer for this key comment that led to the following major changes in the revised version:

- We have extended our methods to identify ARs that preceded and were synchronized with heavy precipitation events in the region delimited by the red box in Figure 5. A new subsection describing this filtering of ARs has been added to the Methods section.
- To calculate the composites shown in figure 6, we now only use 40 ARs, those that were synchronized with heavy precipitation events of most grid cells in the red box.
- We have also prepared a figure showing the synoptic conditions for the days following the landfall of ARs that were not synchronized with heavy precipitation events in the red box. We describe and discuss this result in the Appendix.
- Finally, we have prepared a figure showing the synoptic conditions for the days preceding heavy precipitation events in the red box that occurred in the absence of landfalling ARs. We describe and discuss this result in the Appendix.

With these results, we now answer one of the main questions stated by the reviewer. However, we are unable to make a statement regarding the possible inland penetration of ARs that were synchronized with heavy precipitation over Canada. For this, we would need a catalog that tracks the ARs as they penetrate the continent. Unfortunately, the SIO-R1 catalog

only tracks ARs from the coastline backwards to the ocean, so we don't have enough information to answer this question. We also consider that the implementation of a new AR tracking algorithm, the development of a new catalog for North America, or the use of a different AR catalog to answer this question could be considered for future work.

**Specific comments:**

**Title:** The word 'triggered' may not be suitable here as the local triggers of precipitation are not identified in the study. The focus is also on distant rainfall events, well beyond the landfall location (specifically in Canada). I would recommend rewording this, one possible option may be: 'Spatio-temporal synchronization of heavy precipitation in central Canada and landfalling North Pacific atmospheric rivers'

R/ Thanks for your suggestion. We have changed the title to "The role of atmospheric rivers in the distribution of synchronized heavy rainfall over North America".

**Line 6:** Landfall is usually written without a hyphen.

R/ Fixed.

**Line 7:** The term 'rank' is now being favoured over the 'category' wording, with AR ranks referred to as 'AR1, AR2' rather than 'Cat 1, Cat 2'. This is to avoid confusion with hurricane terminology.

R/ Thanks for this comment. The wording has been changed throughout the whole manuscript.

**Line 10:** 'AR strike' is ambiguous, does this mean landfall?

R/Yes, it means landfall. We have replaced the term 'AR strike' not only in this line but throughout the whole manuscript.

**Line 15:** This final conclusion is very broad and does not reflect the findings of the paper. An alternative is to say that this work will lead to a better understanding of inland precipitation events and how changing climate dynamics may impact precipitation occurrence and consequent impacts in a changing climate.

R/ Thanks for your suggestion. The conclusion has been changed accordingly.

**Line 19:** 'where they landfall and cause copious rainfall' is too casual, could be improved to 'and can cause substantial precipitation following landfall'.

R/ Done.

**Line 19 and 21:** Do not need the parentheses inside the parentheses (this occurs multiple times throughout the manuscript).

R/ Thanks for noticing this. It has been corrected throughout the whole manuscript.

**Line 22:** The grammar of this sentence needs to be fixed, too many conjunctions and becomes hard to follow.

R/ We agree. The sentence has been rewritten.

**Line 23:** Need more introduction/literature discussion around increasing water vapor and ARs in a future climate. More nuanced than this sentence suggests.

R/ We agree and have further developed this topic.

**Line 30:** 'Will form at the front of a mid-latitude cyclone…', the term 'front' has strong connotations in atmospheric science and makes this sentence confusing. ARs tend to form as part of the cold front of a mid-latitude cyclone, specifically the pre-cold frontal lower-level jet.

R/ The sentence has been changed.

**Lines 32-39:** This literature is not directly related to the study presented. This introduction should focus more on the inland penetration of moisture (ARs) and AR lifecycle pathways in North America (i.e. Guan and Waliser 2019; https://doi.org/10.1029/2019JD031205).

R/ We agree. The introduction has been rewritten.

**Line 67:** Reanalysis precipitation can be problematic and contain biases, it would be beneficial to acknowledge this and if possible, provide a reference to the accuracy of this data product for the region of interest.

R/ A new paragraph on this topic has been included.

**Methods:** Very well written methods section, clear and concise. Could possibly use more description of the network analysis to make this more accessible, specifically for those unfamiliar with this approach.

R/ Further details have been added to the methods section. We hope this makes the methodology more accessible to a broader audience.

**Lines 154-156:** These sentences need rewording.

R/ We agree. We have rewritten this part to clearly explain why we did the analysis considering only ARs of level AR3 or higher. We have also added a new figure to the appendix showing ES for different AR levels to better illustrate this point.

**Figure 2:** The spacing between precipitation on the coast and central Canada is intriguing. Is there a suitable reason for this? Possibly the role of topography?

R/ We have checked if the topography is a driver for this connection, or if it triggers the precipitation over Canada by lifting the moisture transported there after the landfall of ARs. To do that, we used a DEM and calculated the topographic gradient of each pixel of the study region. Besides the topography of the Rocky Mountains, no significant gradient was found further inland, so we discarded this hypothesis. The most suitable reason for the spacing between precipitation on the coast and central Canada is the distance between these 2 areas, which is on the order of thousands of kilometers. We hypothesize that landfalling ARs act as a source of moisture for synchronized heavy precipitation over Canada. Then, the moisture needs to be transported over this long distance and this temporal spacing is expected.

**Line 188:** It appears the text has the sign wrong than what the graph shows, with negative divergence values at the coast and positive divergence values out in the Pacific.

R/ You are right, we have corrected this.

**Line 197:** Double parentheses used again, these can be removed.

R/ Fixed.

**Line 220:** The word 'cascade' makes it sound like many events, from what I interpret these results identify a sequence of 2 rainfall events, Coastal and then central Canada with 12 days following.

R/ We consider that the use of the word 'cascade' is correct in this context. Cascading events are unforeseen chains of dependent phenomena due to an originating event or triggering hazard. In this case, the triggering hazard is the land-falling AR and the dependent phenomena are the heavy precipitation events that originate on the coast and serve as sources for heavy precipitation events over Canada. You are right, these events are also sequential, we were able to identify their timeline by setting the parameters of ES and running the complex network analysis, but more important than a sequence, they are a cascade: sequential but also dependent.

One last remark we would like to make in this regard is that our results are not for 2 rainfall events, but for all the heavy rainfall events, during a period of 39 years, over the entire study region. Please recall that, when we calculate ES to set up a complex network, we are associating the heavy precipitation events of each grid cell, during the whole time series, with the heavy precipitation events of all the other grid cells in the study region, during the entire study period. So, our results show that, between 1979 and 2018, there has always been a significant synchronization between the heavy rainfall events of these 2 areas. The coast has been acting as a source and Canada has been acting as a sink for this synchronized but delayed pattern of heavy precipitation, therefore, we consider it correct to call this phenomenon a cascade.

**Line 226-227:** The grammar of this sentence needs improving, hard to follow.

    R/ We agree. The sentence has been rewritten.

**Section 3.5:** This section is an assessment of the synoptic conditions and not the climate.

    R/ You are right. We have changed the section title.

**Line 230-231:** 'The moisture is distributed further to the mainland', this is difficult to see on the figure, particularly over Canada. It appears that the moisture flux remains relatively in the same broad location, but becomes weaker.

    R/ You are right. With the new methodology to identify synchronized ARs, this effect is now clearly visible.

**Lines 232-234:** This interpretation is problematic. The authors appear to be referring to maximums in geopotential height anomaly as 'the cyclonic storm', at 500 hPa this maximum rather signifies a ridge with a trough in the northwest of the scene (not a high-pressure area). This mid-level pressure dipole presented here implies a southwesterly geostrophic wind, bring warm moist air into the northern regions of North America. Since the vast majority of moisture transport is also in the lower atmosphere, the surface low (cyclone) is the key feature of interest, which will be located more towards the 500 hPa minimum in the northwest, with an assumed cold front running from the lower left of the scene towards the Canadian coastline which facilities the moisture transport. These results appear to show the importance of ridging over the Western USA for strong ARs to make landfall in Canada. I would recommend a full rewrite of this section to ensure the interpretation of this figure is correct.

    R/ Thank you for this very specific and detailed comment. To overcome this pitfall, we have extended our methodology, produced new figures, and rewritten this section.

**Line 237:** The word 'continent' after Pacific is not required.

    R/ The word continent is in parentheses because it belongs to the word southward. We use these parentheses to avoid a long and repetitive sentence.

**Figure 6:** This figure presenting the composite during all high ranked ARs. It will be very interesting if the authors could make similar figures for the conditions of when the landfalling AR does and does not cause synchronized precipitation in Canada. Are there specific atmospheric conditions that don't allow for the deep inland penetration of moisture and subsequent precipitation? This would be a great appendix figure.

    R/ We agree and appreciate this suggestion, which has helped us to significantly improve our results and discussion. Please find these new figures in the Appendix.

**Line 260:** This sentence is not easy to read, possible grammatical error, '…with ARs synchronize initially…', this is the confusing part. May need to reword this.

R/ The sentence has been rewritten.

**Line 270:** '…Maintaining a moisture flux…' of what magnitude? Over 250 kg m$^{-1}$ s$^{-1}$?

R/ To calculate the IVT anomaly, we didn't threshold the magnitude of this variable but used every value greater than 0. Therefore, we can not state that the moisture flux is over 250 kg m$^{-1}$ s$^{-1}$. We can only state that there is an anomalous moisture influx above the average conditions. The sentence has been rewritten accordingly.

**Line 280:** Similar to comments about the abstract, this sentence needs to be reworded to better reflect the benefit of the science presented in this paper.

R/ The sentence has been rewritten.

**Reviewer 2.**

The authors apply concepts from complex networks to study Atmospheric Rivers. This group is one of the leading groups in applying network to climate. The application is novel, the presentation is very good, and the results may open new ways to understand these phenomena.

I read the comments by RC1 and I agree with his/her suggestions to improve the discussion. I urge the authors to consider these comments.

R/ Thanks for your comment. Please refer to our reply to Reviewer 1.

---

## Author Response (AR2)

**THE ROLE OF ATMOSPHERIC RIVERS IN THE DISTRIBUTION OF HEAVY PRECIPITATION EVENTS OVER NORTH AMERICA**

**Response to reviewer's comments**

Sara M. Vallejo-Bernal

April 2023

This document contains responses to the review of the manuscript "The role of atmospheric rivers in the distribution of heavy precipitation events over North America". It is an attachment to the revised version of the manuscript submitted to the Hydrology and Earth System Sciences Journal.

Every comment was addressed and is listed below along its corresponding response. Following the recommendations of the reviewer, minor changes were done. Additionally, a new figure was included in the supplementary material, Fig. S6 was corrected, and some final refinements were made in the Methods section. I would like to sincerely thank the reviewer for the work and effort put into evaluating this manuscript.

**Reviewer 1.**

**Minor corrections:**

**Line 41-42:** The grammar of this sentence need correcting, particularly the phrase '...provide nowadays the guidelines...'. I think by removing the words 'nowadays the', the sentence becomes much easier to read.

> R/ We agree, these words were removed.

**Line 90:** Change '...were ERA5' to '...where ERA5'

> R/ Done.

**Line 93:** Good inclusion of errors associated with the data and reasoning why you have used it. I think you have left off the most important reason to use ERA5 precipitation - ERA5 provides a globally gridded, hourly precipitation product. You could adjust this sentence so that at the end it says '...and ultimately because it provides a globally gridded, hourly precipitation product.'

> R/ The sentence has been adjusted.

**Line 296-258:** This wave pattern is typical of a Rossby wave train which could certainly contain previous or synchronized ARs and HPE propagating eastward. Your suspicions are valid, but this could be made more general by stating that - the wave pattern resembles a large scale midlatitude wave train that may indeed contain alternating moist and dry advection which may include a sequence of landfalling AR-HPE synchronization. Also the phrase 'climate features', isn't quite accurate here since you are considering synoptic time-scales. Instead, it could just say 'We suspect that this wave pattern resembles...'

R/ Thanks for this suggestion. We have changed this sentence.